# Microfluidic-spinning construction of black-phosphorus-hybrid microfibres for non-woven fabrics toward a high energy density flexible supercapacitor

Xingjiang Wu[1], Yijun Xu[2], Ying Hu[3], Guan Wu [1], Hengyang Cheng[1], Qiang Yu[2], Kai Zhang[2], Wei Chen [2] & Su Chen [1]

Flexible supercapacitors have recently attracted intense interest. However, achieving high energy density via practical materials and synthetic techniques is a major challenge. Here, we develop a hetero-structured material made of black phosphorous that is chemically bridged with carbon nanotubes. Using a microfluidic-spinning technique, the hybrid black phosphorous–carbon nanotubes are further assembled into non-woven fibre fabrics that deliver high performance as supercapacitor electrodes. The flexible supercapacitor exhibits high energy density (96.5 mW h cm$^{-3}$), large volumetric capacitance (308.7 F cm$^{-3}$), long cycle stability and durability upon deformation. The key to performance lies in the open two-dimensional structure of the black phosphorous/carbon nanotubes, plentiful channels (pores <1 nm), enhanced conduction, and mechanical stability as well as fast ion transport and ion flooding. Benefiting from this design, high-energy flexible supercapacitors can power various electronics (e.g., light emitting diodes, smart watches and displays). Such designs may guide the development of next-generation wearable electronics.

[1] State Key Laboratory of Materials-Oriented Chemical Engineering, College of Chemical Engineering, Jiangsu Key Laboratory of Fine Chemicals and Functional Polymer Materials, Nanjing Tech University, Nanjing 210009, PR China. [2] i-Lab, Suzhou Institute of Nano-tech and Nano-bionics, Chinese Academy of Sciences, Suzhou 215123, PR China. [3] Institute of Industry and Equipment Technology, Hefei University of Technology, Hefei 230009 Anhui, PR China. Correspondence and requests for materials should be addressed to G.W. (email: gwu2016@njtech.edu.cn) or to K.Z. (email: kzhang2015@sinano.ac.cn) or to S.C. (email: chensu@njtech.edu.cn)

New energy storage technologies attract attention from fundamental science researchers as well as industrial communities[1,2]. The development of efficient energy storage capabilities by using advanced materials is a key area of interest. In this context, flexible energy storage devices[3–10], e.g. flexible batteries[11], supercapacitors (SCs)[12], solar cells[13] and fuel cells[14], are highly desirable because of their widespread potential applications in roll-up displays, implantable medical devices, wearable sensors, smart robots, etc.[9,12,15,16]. In particular, flexible SCs offer high power density, fast charge/discharge rates and long cycling life[17–23]. However, two major challenges still exist: (1) relatively low energy density and rate stability hinder scientific research and practical applications[12,24], and (2) a lack of large-scale technologies for fabrication of electrodes with outstanding mechanical strength limits the development of flexible and deformable SCs[25,26]. To this end, research is focused on developing advanced structural materials and synthesis technologies for flexible electrode.

Two-dimensional (2D) layered nanomaterials[26–28] with atomically thin structures, low framework density, and high electrical and electrochemical performances, have emerged as promising candidates. In particular, the discovery of graphene with its 2D structure and large specific surface area (SSA) has motivated intensive studies of 2D materials for flexible SCs with high energy density. For instance, the specific capacitance and energy density of graphene[29] are 3.05 F cm$^{-3}$ and 2.1 mW h cm$^{-3}$, respectively, and the energy density of graphene-based hybrid carbon materials[17] could reach 6.3 mW h cm$^{-3}$. In addition, next-generation 2D nanomaterials, such as graphitic carbon nitride[30], transition metal oxides[31], transition metal dichalcogenides[32] and MXenes[33] have been intensely explored. For example, an electrochemically active molybdenum disulfide (MoS$_2$)-based composite electrode[34] exhibited an enhanced specific capacitance of 6.1 F cm$^{-3}$. An electrostatic-assembled MXene-based composite electrode[35] displayed an ultra-high energy density of 32.6 mW h cm$^{-3}$, benefiting from improved ion surface accessibility and rapid diffusion.

To enable practical application, fabrication technologies for flexible electrodes with small volume, flexibility and deformability are highly needed. Much effort has been expended on coating deposition[36], on-chip writing[29], casting[25], wet-spinning[18] and dry-spinning[37], allowing materials to be well dispersed in end products. Alternative approaches include incorporation of heteroatoms[25,38,39] (e.g. N, S, P), metal oxides[40,41] (MnO$_2$, RuO$_2$, Co$_3$O$_4$) and conducting polymers[42,43] such as polyaniline (PANI) and polypyrrole (PPy) to promote energy storage capability through quantum and pseudo-capacitance. In most cases, however, uncontrollable poor miscibility during complex fabrication tends to limit the achievable energy density, interface charge (ion) transfer, conductivity of the electrode and mechanical strength, which motivates us to find a suitable solution.

Herein, we develop a new type of flexible electrode based on a hetero-structured black phosphorous (BP)–carbon nanotubes (CNTs) hybrid that presents one of the highest energy densities (96.5 mW h cm$^{-3}$) compared with previously reported 2D nanomaterials (MoS$_2$-based[34], 1.6 mW h cm$^{-3}$; graphene-based[17], 6.3 mW h cm$^{-3}$; MnO$_2$-based[22], 11.1 mW h cm$^{-3}$ and MXene-based[35], 32.6 mW h cm$^{-3}$). As a new class of 2D nanomaterials, BP possesses a unique lamellar structure, high carrier mobility, and electrical and optical anisotropy[44–46], which has been applied in lithium/sodium ion batteries[47,48], electrocatalysis of oxygen evolution[49], capacitors[50] and photodetectors[51]. Furthermore, BP has a larger interlayer spacing (5.3 Å) and a weaker van der Waals interlayer interaction than graphene (3.6 Å)[52]; thus, intercalation of ions such as Li$^+$ or Na$^+$ is easier[52,53]. Leveraging those unique properties, Liu et al.[54] used physical

sonication to fabricate BP–CNTs hybrid materials that instilled the SC with excellent electrochemical performance, including a specific capacitance of 41.1 F cm$^{-3}$ and energy density of 5.71 mW h cm$^{-3}$. However, poor electron conduction and stability between the BP interlayers has limited further improvements in the energy storage capability.

In this work, considering the chemical bonding design, we propose a hetero-structured BP/CNTs hybrid materials where one-dimensional (1D) nanowire (CNTs) are chemically bridged within a 2D nanosheet (BP) via a P–C bond connection under a high-heat treatment. In such an architecture, BP shows graphene-like conductivity. The CNTs are embedded in situ in the BP flakes, promoting lamellar electron conduction, enhanced mechanical stability and alleviating layer restacking of the BP nanoflakes; thus, the conductive networks are produced with ionic channels for fast ion diffusion and ion flooding. In addition, we applied a microfluidic-spinning-technique (MST)[55,56] to fabricate BP/CNTs-based microfibres (>50 m) and further assemble the fibres into a flexible non-woven fabric at a large-scale (>7 cm). The as-prepared fabric electrodes have outstanding mechanical strength (Young's modulus: 313 MPa; break elongation: 17.96%), electrical conductivity, flexibility and deformability. Owing to those advantages, flexible SCs exhibit high energy density, large specific capacitance, long-life cycling stability and durability upon deformation, which can successfully power various electronics, including LEDs, a smart watch and displays. This finding highlights the importance of 2D/1D hetero-structures towards high-performance energy storage and advances the use of MST to guide the design of power sources for wearable electronics.

## Results

**Synthesis of hetero-structured materials**. For SCs, the key design task for the electrodes is to build a stable network structure that facilitates electron conduction and ion diffusion[57,58]. In this regard, we designed hetero-structured BP/CNTs materials, as illustrated in Fig. 1a and Supplementary Figure 1. By high-heat treatment of a mixture of precursors, red phosphorus was evaporated and transferred to a BP layered structure on the surface of CNTs, thereby generating hierarchically hetero-structured BP–CNTs crystals. Due to the in situ reaction, chemically bridged BP–CNTs with a porous network are formed. As expected, while maintaining the special 2D layered structure, the network also exhibits improved lamellar electrical conductivity and mechanical stability because of the insertion of CNTs, which are beneficial for achieving both high energy and power densities. To establish the stable operation of BP–CNTs, chemically passivated modification was performed by using 4-nitrobenzene diazonium (4-NBD)[59]. As a result, BP–CNTs retained excellent stability when stored in air for more than 1 month (Supplementary Figure 2).

To make the materials flexible and wearable, MST, which is an easy-to-perform process with organized morphologies and compositions[55,60,61], is presented. As shown in Fig. 1b, a triphase microfluidic device featuring one core flow and two sheath flows is developed to allow three liquid phases to flow, coagulate and generate microfibres. Next, microfibres are interfused and interconnected with each other, forming non-woven fibre fabrics because of residual-solvent-caused heat-welding[62]. To further improve the volumetric density, the fabrics are mechanically compressed to compact into free-standing films that are tailored into various shapes. Figure 1c shows the construction of a flexible SC. Specifically, a half-dried poly(vinylidene fluoride-co-hexafluoropropylene) (PVDF-HFP)/1-ethyl-3-methylimidazolium tetrafluoroborate (EMIBF$_4$) ion liquid electrolyte layer is laminated with two conductive fabric film layers via a hot-press process. Intriguingly, the SCs are flexible enough to withstand a

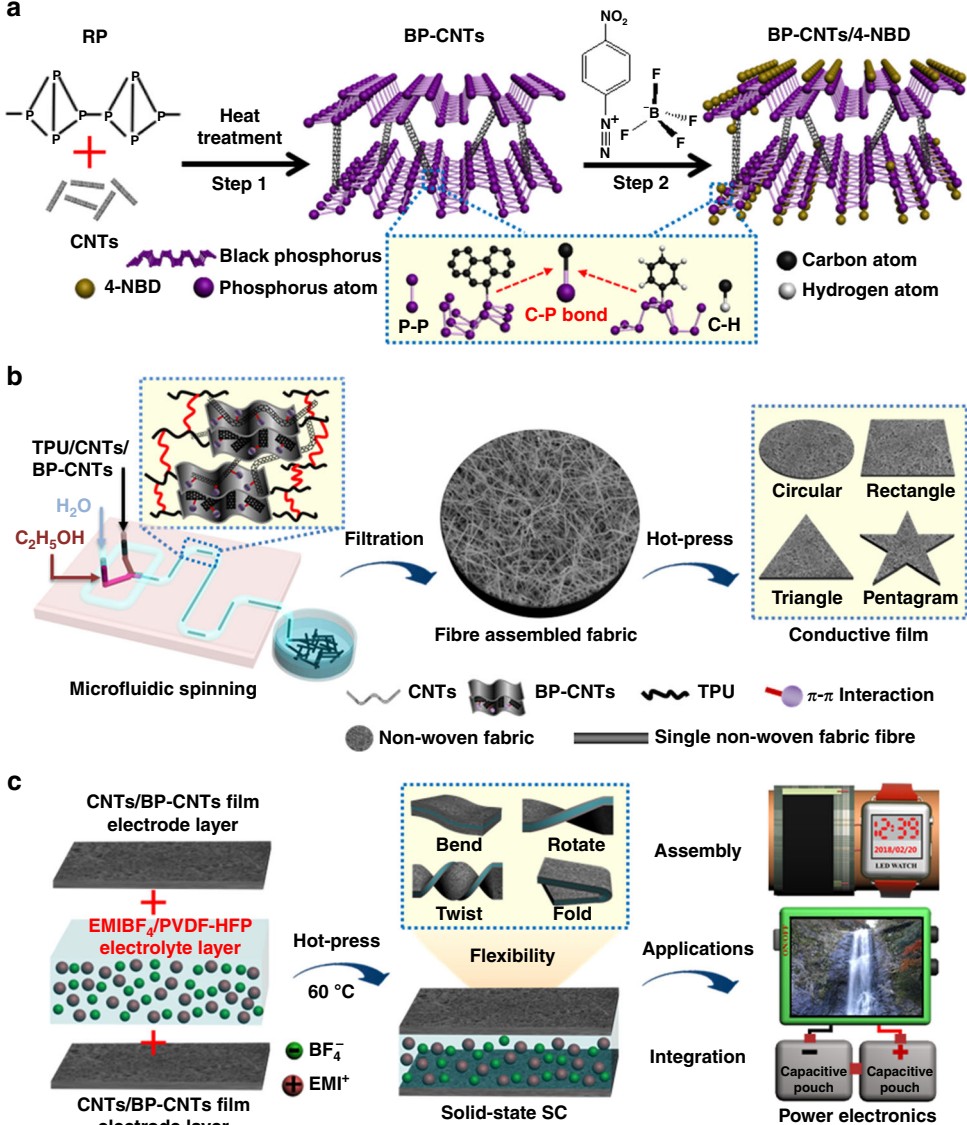

**Fig. 1** Schematic illustrations. **a** Synthesis of black phosphorous chemical-bridged carbon nanotubes (BP–CNTs) by heat treatment of black phosphorous (BP) and carbon nanotubes (CNTs), and chemical passivation of BP–CNTs using 4-nitrobenzene diazonium (4-NBD). Red phosphorous is abbreviated as RP. **b** Microfluidic-spinning-technique (MST) fabrication of microfibres via a triphase microfluidic device consisting of one core flow and two sheath flows and fibres assembled into non-woven fabrics, which can be cut into different shapes. Thermoplastic polyurethane is abbreviated as TPU. $C_2H_5OH$ is ethanol. **c** Construction of flexible supercapacitor (SC) by hot pressing of two conductive fabric layers and one polymer-supported ionic liquid electrolyte layer. The supercapacitors have the potential to power various electronics for application. $EMI^+$ is 1-ethyl-3-methylimidazolium, $BF_4^-$ is tetrafluoroborate, $EMIBF_4$ is 1-ethyl-3-methylimidazolium tetrafluoroborate, and PVDF-HFP is poly(vinylidene fluoride-co-hexafluoropropylene). Guan Wu is the creator of the waterfall photo in the powered electronic device

variety of deformations, and can power LEDs, smart watches and displays.

**Structural characterization of hetero-structured materials**. The microstructural morphologies of the as-prepared BP–CNTs crystals are characterized by transmission electron microscopy (TEM) and scanning electron microscopy (SEM). Figure 2a shows the hybrid material exhibits a sheet/wire structure of CNTs embedded within the BP. Moreover, shown at higher magnification (Fig. 2b), CNTs are notably well-coupled with BP flakes. To elucidate the crystal structure of BP–CNTs, high-resolution TEM (HRTEM) is performed (Fig. 2c). As expected, a lattice fringe with an interplanar spacing of 0.26 nm is obtained that corresponds to the (040) lattice plane of the BP crystal. The

surrounded lattice spacing of 0.33 nm is equivalent to the (002) plane of the CNTs. As collected from the selected zone (Fig. 2d), a well-dispersed spatial distribution of C (Fig. 2e) and P (Fig. 2f) elements is revealed by the energy dispersive X-ray spectroscopy (EDS) mapping images. The clear mapping images of elements verify the successfully achieved uniformity of the BP–CNTs material. SEM images (Fig. 2g, h) also show that the interlayer of BP is satisfactory by cross-linked CNTs, creating a stable and hierarchical porous network (Fig. 2i). The CNTs interposed between BP layers not only prevent the self-restacking of BP nanosheets with developed porosity but also improve the lamellar conductivity.

X-ray photoelectron spectroscopy (XPS) is performed to survey the typical C 1s and P 2p peaks, characterizing the formation of P–C bonds in BP–CNTs[48,53,63]. Obviously, the BP–CNTs show

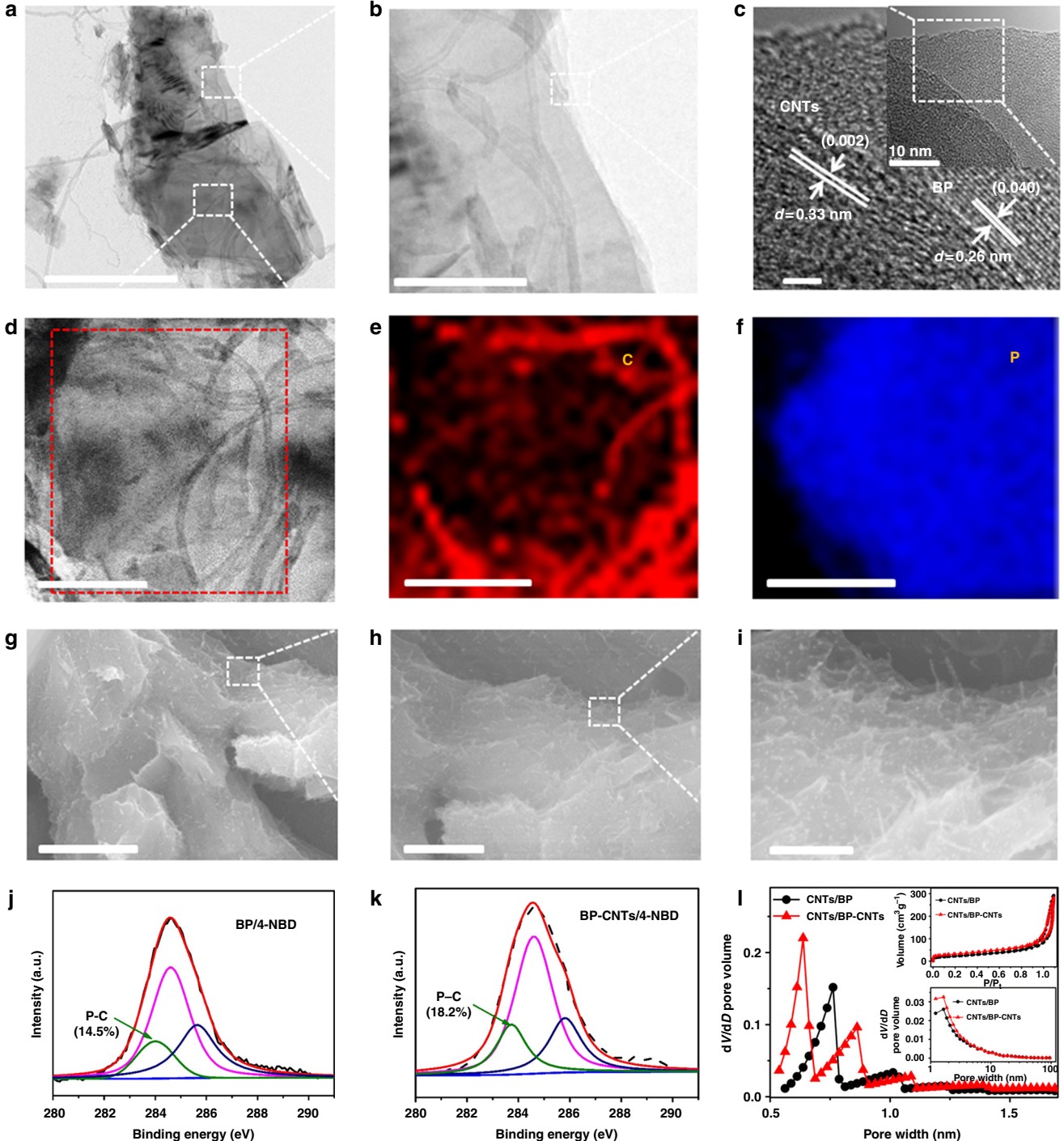

**Fig. 2** Structural characterization. **a**, **b** Transmission electron microscopy (TEM) images of black phosphorous chemical-bridged carbon nanotubes (BP–CNTs) at low and high magnifications, respectively. Scale bar: **a** 500 nm; **b** 100 nm. **c** High-resolution TEM (HRTEM) image of BP–CNTs, scale bar: 2 nm. **d** High magnification of the selected domain in **a**, scale bar: 100 nm. The corresponding energy dispersive X-ray spectroscopy (EDS) elemental mapping images of C (**e**) and P (**f**), scale bar: **e** 100 nm; **f** 100 nm. Surface scanning electron microscopy (SEM) images of BP–CNTs at low (**g**) and high magnifications (**h**), scale bar: **g** 2 μm; **h** 1 μm. **i** Structural illustration of BP–CNTs, scale bar: 400 nm. High-resolution C 1s X-ray photoelectron spectroscopy (XPS) spectra of black phosphorous (BP) (**j**) and BP–CNTs (**k**). **l** Micro-pore size distributions. Insets are typical nitrogen adsorption and desorption isotherms and meso-pore size distributions. 4-NBD is 4-nitrobenzene diazonium

P–C formation at ~284 eV binding energy in the C 1s spectra in Fig. 2j, k, confirming the chemical bridge between BP and CNTs under high temperature treatment. Supplementary Figure 3 further investigates the modification of 4-NBD with BP–CNTs[59].

According to Supplementary Figure 3a and b, the BP/4-NBD shows P–C formation with 14.5% at ~284 eV binding energy in

the C 1s spectra. The P–C intensity of the BP–CNTs exhibits an enhancement of 18.2%, which verifies the 4-NBD modification of BP–CNTs. These intensity levels of P–C bonds are also enhanced as observed in P 2p spectra (Supplementary Figure 3c and d). As revealed by the Raman spectra (Supplementary Figure 4), the CNTs/BP mixture exhibits three peaks at 362, 438 and 466 cm$^{-1}$,

corresponding to the $A_{1g}$, $B_{2g}$ and $A_{2g}$ modes of BP[48,63–65], respectively. The two strong peaks located at 1352 and 1591 cm$^{-1}$ are assigned to the D and G bands of the CNTs, respectively. However, it is worth noting that a small peak at 703 cm$^{-1}$ in CNTs/BP–CNTs is attributed to the P–C bond[53], revealing that a small portion of P–C bonds are formed in the BP–CNTs. The whole pore size distributions and their contributions to the SSA from micro- to meso- to macro-pores are illustrated in Fig. 2l and Supplementary Table 1. Because the flexible electrode materials are mixed with pure CNTs, the pore size measurements include both active materials and pure CNTs. The CNTs/BP material has an average pore diameter of 6.28 nm with a sharp peak at 0.76 nm, which shows a pore size distribution from 2.4 to 190.5 nm. However, the CNTs/BP–CNTs material generates more micro/ meso-pores and maintains a smaller mean pore diameter of 5.27 nm with two sharp peaks at 0.64 and 0.86 nm. Due to these major micro/meso-pores, CNTs/BP–CNTs have a relatively larger SSA (109.46 m$^2$ g$^{-1}$) than that of CNTs/BP (70.19 m$^2$ g$^{-1}$). The overall characterization confirms the existence of stable chemically bridged hetero-structured BP–CNTs with more ion accessible channels, providing a unique layered structure, electron conduction, electrochemical activity and mechanical reinforcement.

**Construction of fabric electrode and performance of supercapacitor.** The second set of experiments was focused on MST for fabrication of flexible electrodes (Supplementary Figure 5). As illustrated in Fig. 3a, the microreactor with three microchannels consists of one core flow (well dispersion of 4-NBD modified BP–CNTs, CNTs and TPU), and two sheath flows ($C_2H_5OH$ and $H_2O$). Typically, the core flow of the dispersion (flow rate: 70 mL h$^{-1}$) was injected into the microreactor by syringe, followed by pumping the sheath flow of $C_2H_5OH$ (flow rate: 55 mL h$^{-1}$) into the device for pre-coagulation of dispersion. Then, the dispersion was deeply coagulated to form microfibre by extruding water (flow rate, 40 mL h$^{-1}$) through solvent exchange. Subsequently, the as-prepared fibres were filtrated, dried and hot-pressed. It is because the solvent–vapour-mediation allows multiple junctions of fibres so that fibres could be interfused and interconnected with each other, forming non-woven fibre fabrics[62,66,67]. As shown in the SEM image in the inset of Fig. 3a, the fabric consists of uniform fibres with a diameter of about approximately 80 μm. To improve the mechanical properties, fabrics were compressed to form compact films with excellent electrical conductivity (75.2 Ω m$^{-2}$) and mechanical strength (Young's modulus: 313 MPa; break elongation: 17.96%) (Supplementary Figure 6). It is worth mentioning that the mechanical elongation of fabric electrodes by MST is much higher than that of other literatures reported electrode materials (graphene chiral liquid crystals fibres: 5.8%[68]; graphene fibres: 6%[69]; CNTs/ZIF-8: 6.42%[25]; RGO +CNTs@CMC: 10%[18]), confirming the excellent flexibility and deformability of the SCs. The compact films with notably high mechanical strength can be cut into various shapes and operated with different deformations, highlighting the robustness and flexibility of the SCs.

The energy density of SCs is proportional to the capacitance and the square of the applied potential[6]. In addition to an electrode material and structure design, an electrolyte layer is desirable. To this end, we used EMIBF$_4$/PVDF-HFP as a solid-state electrolyte layer owing to its relatively wide potential window (~0–3 V) and operating stability in air[30]. In our system, we designed three kinds of structural electrodes: a wire/wire structure of pure CNTs (Fig. 3b), a wire/sheet structure of CNTs/ BP (Fig. 3c), a wire/sheet/wire hetero-structure of CNTs/ BP–CNTs (Fig. 3d). In the main electrochemical testing, the BP

and BP–CNTs used in fibre-based fabrics are all modified by 4-NBD; materials unmodified by 4-NBD are also considered for comparison. Cyclic voltammetry (CV) and galvanostatic charge/ discharge measurements are implemented to evaluate the electrochemical performance. As shown in Fig. 3e, pure CNTs present a small rectangular shape of electric double-layer capacitance (EDLC), and a strongly enlarged CV curve of CNTs/BP with redox reaction reveals that the developed pseudocapacitance is derived from the BP. Furthermore, CNTs/ BP–CNTs display the largest CV area, indicating the best energy storage ability and ion diffusion throughout SCs. CNTs/ BP–CNTs also exhibit the best charge/discharge behaviour with a symmetric triangular shape, implying the high reversibility of the device (Fig. 3f, Supplementary Figures 7, 8 and 9). Figure 3g shows the specific volumetric capacitances of SCs. It is found that CNTs/BP–CNTs display the highest specific capacitance of 308.7 F cm$^{-3}$ at 0.1 A cm$^{-3}$, whereas that of CNTs/BP and pure CNTs are 235.5 and 119.6 F cm$^{-3}$, respectively. This outstanding result is expected to originate from the unique 2D/1D hetero-structure, in which BP–CNTs maintain an enhanced interlayer conductivity and electrochemical activity. To the best of our knowledge, this capacitive level of our SC exceeds that of previously reported nanomaterial electrode-based flexible SCs[17,29,34,70–74].

To investigate how the unique designed structure affects ion diffusion and storage in SCs, electrochemical impedance spectroscopy (EIS) measurement is conducted[75,76]. Figure 3h shows the Nyquist plots as simulated by the inserted equivalent circuit model, clarifying the impedance of a single depressed semicircle at high frequencies, diffusion (Warburg impedance) at medium frequencies, and intercalation capacitance (structural feature) at low frequencies. Based on the fitting analysis in Supplementary Table 2, the inner resistances ($R_0$) of SCs are at nearly the same levels, implying that SCs have been constructed and measured in the same condition. When considering the contact impedance ($C_1/R_1$), pure CNTs (0.32 mF/0.71 Ω) have a value smaller than that of CNTs/BP (0.28 mF/1.22 Ω), indicating that the added active BP reduces the electric conductivity. However, when bridging CNTs within the BP, the electron conduction of CNTs/ BP–CNTs (0.31 mF/0.83 Ω) is enhanced to the same value as that of pure CNTs, confirming that electron conduction in the BP–CNTs network is promoted by CNTs insertion. Regarding ion diffusion ($Z_w$), CNTs/BP–CNTs exhibit a lower value (23.75 Ω) than those of CNTs/BP (38.34 Ω) and pure CNTs (49.27 Ω). Supplementary Figure 10 shows the phase angle dependence on the frequency for SCs. For the CNTs/BP–CNTs SC, the phase angle is close to −90° at low frequencies, indicating a better ideal capacitive behaviour than that of CNTs/BP and pure CNTs. The characteristic frequency $f_0$ at a phase angle of −45° or its corresponding relaxation time, also called the RC time constant ($\tau_0 = 1/f_0$) marks the point where the resistive and capacitive impedances are equal[6,77]. The CNTs/BP–CNTs, CNTs/BP and CNTs exhibit an $f_0$ of 0.202, 0.14 and 0.107 Hz, respectively, corresponding to RC time constants of 4.95, 7.14 and 9.35 s, respectively. The faster frequency response of CNTs/BP–CNTs indicates an enhanced ion transport rate within the hetero-structured electrodes. As a result, the intercalation capacitance ($C_2$) of the CNTs/BP–CNTs (2.54 F) is larger than that of CNTs/ BP (1.62 F) and pure CNTs (0.79 F). The improvement is attributed to the BP–CNTs designed structure, in which 2D nanosheets provide a favourable path channel for ion transportation and the embedded CNTs enhances layer–layer conductivity. The resulting high energy storage performance demonstrates the crucial importance of the architected hetero-structured BP–CNTs framework for ion faster diffusion and greater accessibility.

The life cycle stabilities of SCs are examined by continually testing the charge/discharge process. As shown in Fig. 3i, pure

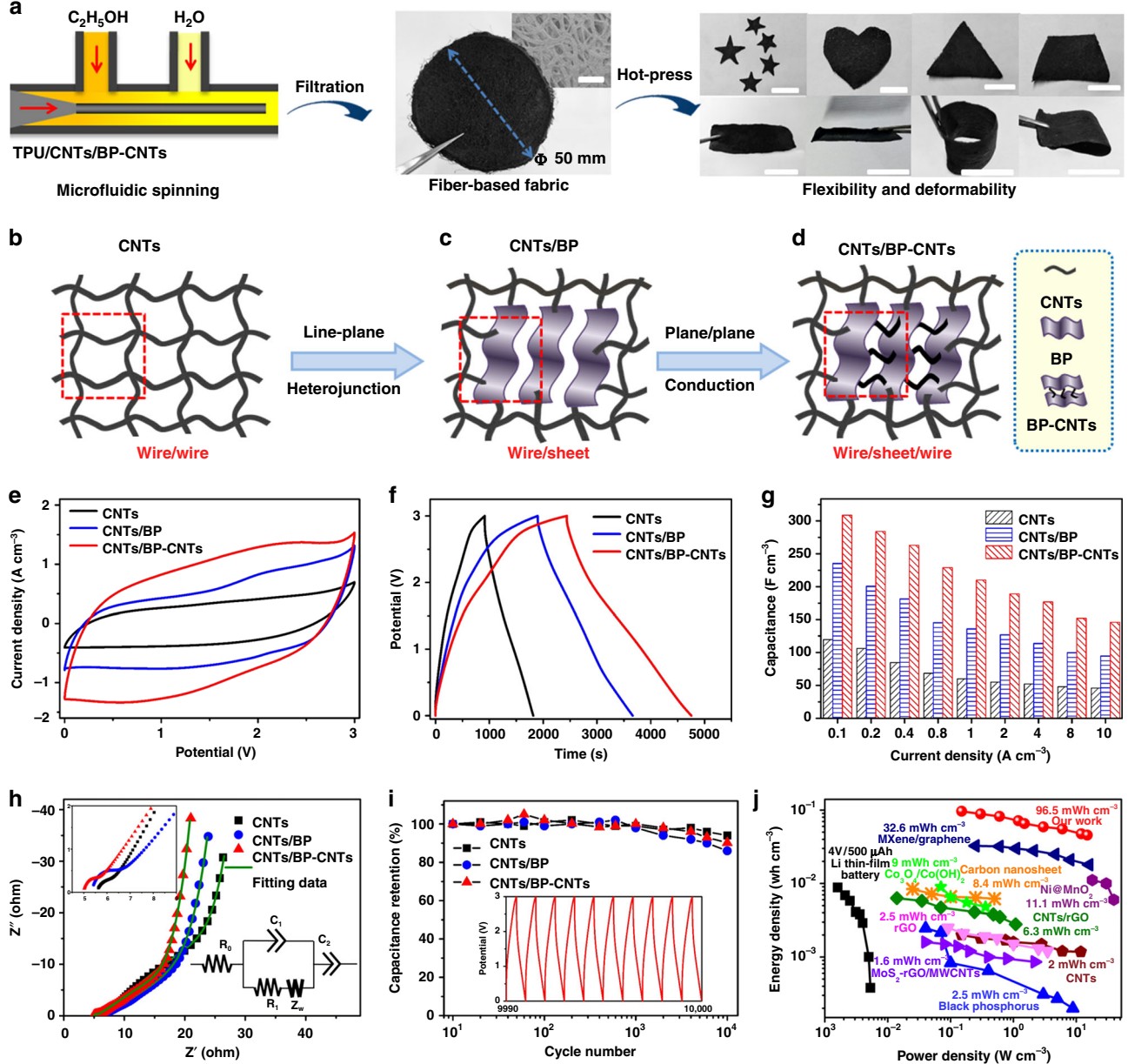

**Fig. 3** Electrochemical performance of flexible supercapacitors. **a** Microfluidic-spinning-technique (MST) fabrication of fibres-based non-woven fabrics. Left part: illustration of the device microchannel. Middle part: photograph of non-woven fabric. The inset: scanning electron microscopy (SEM) image of the fabric, scale bar: 500 μm. Right part: non-woven fabric that can be cut into various shapes and subjected to various deformations. All scale bars in photos are 2 cm. **b–d** Schematic illustration of the designed electrode structure. Carbon nanotubes is abbreviated as CNTs. BP is defined as black phosphorous. CNTs/BP refers to the physical mixture of carbon nanotubes and black phosphorous, CNTs/BP–CNTs refers to the physical mixture of carbon nanotubes and carbon nanotubes chemical-bridged black phosphorous. **e** Cyclic voltammetry (CV) curves of supercapacitors (SCs) at a scan rate of 10 mV s$^{-1}$. **f** Galvanostatic charge/discharge curves at a current density of 0.1 A cm$^{-3}$. **g** Calculated specific capacitances under different current densities. **h** Electrochemical impedance spectroscopy (EIS) analysis of SCs. The inset figures show the depressed semicircle of the Nyquist plots and the equivalent circuit model. The symbols denote experimental data, while the green lines represent the fitted data. **i** Cyclic testing of SCs under a voltage of 3 V at a current density of 0.4 A cm$^{-3}$; inset: galvanostatic charge/discharge curves after 10,000 cycles. **j** Energy density versus power density of SCs compared with other electrode-based energy storage systems. The BP and BP–CNTs are all modified by 4-nitrobenzene-diazonium (4-NBD). MoS$_2$-rGO/MWCNT[34] is Molybdenum disulfide-reduced graphene oxide/multi walled carbon nanotubes. CNT[72] is carbon nanotubes. rGO[73] is reduced graphene oxide. Black phosphorous[74]. CNTs/rGO[17] is carbon nanotubes/reduced graphene oxide. MnO$_2$/carbon cloth[70] is manganese dioxide/carbon cloth. Carbon nanosheet[71]. Graphene/PANI[79] is graphene/polyaniline. Co$_3$O$_4$/Co(OH)$_2$[80] is four oxidation of three cobalt/cobalt hydroxide. Ni@MnO$_2$[22] is nickel@manganese dioxide. MXene/Graphene[35] is metal carbides/graphene. Li thin-film battery[3]

CNTs retains 94.2% of initial capacitance after 10,000 cycles. However, the capacitance retention of CNTs/BP is decreased to 86.1% because of the poor interaction between BP layers. Notably, the capacitance retention of CNTs/BP–CNTs is increased to 90.2%, revealing that the embedded CNTs improves mechanical

stability. For SCs, the energy and power densities are more key parameters to evaluate the energy storage performance. Figure 3j shows the CNTs/BP–CNTs have a volumetric energy density of 45.6–96.5 mWh cm$^{-3}$ at a power density of 0.15–15 W cm$^{-3}$. We conclude that both the energy and power densities of our flexible

SC are high. To our knowledge, the maximum volumetric energy density of our SC is one of the highest values among commercial product and similarly composited 2D layered materials (Supplementary Table 3), such as Li thin-film battery, CNTs (2 mWh cm$^{-3}$), rGO (2.5 mWh cm$^{-3}$), BP (2.5 mWh cm$^{-3}$), CNTs/rGO (6.3 mWh cm$^{-3}$), MnO$_2$/carbon cloth (8.3 mWh cm$^{-3}$), carbon nanosheet (8.4 mWh cm$^{-3}$), graphene/PANI (8.80 mWh cm$^{-3}$), Co$_3$O$_4$/Co(OH)$_2$ (9.4 mWh cm$^{-3}$), Ni@MnO$_2$ (11.1 mWh cm$^{-3}$) and MXene/graphene (32.6 mWh cm$^{-3}$).

To investigate the effect of 4-NBD, the performances of CNTs/BP and CNTs/BP–CNTs without modification are compared. As illustrated in Supplementary Figure 11a and Fig. 3f, CNTs/BP and CNTs/BP–CNTs without 4-NBD modification have a lower charge storage performance than electrodes with 4-NBD modification, mainly due to the oxidation of BP, which decreases the electrochemical performance. BP–CNTs and BP–CNTs/CNTs-based SCs without 4-NBD modification present a specific capacitance of 152.3 and 225.1 F cm$^{-3}$, respectively at 0.1 A cm$^{-3}$ (Supplementary Figure 11b), which is smaller than that of 4-NBD modification BP–CNTs (235.5 F cm$^{-3}$) and BP–CNTs/CNTs (308.7 F cm$^{-3}$), as shown in Fig. 3g. Furthermore, the cycle stabilities of BP–CNTs and BP–CNTs/CNTs without 4-NBD modification (Supplementary Figure 11c) decline more prominently relative to that of BP–CNTs and BP–CNTs/CNTs with 4-NBD modification (Fig. 3i). Thus, the 4-NBD modification can improve the stability of SCs. Regarding the energy density, (Supplementary Figure 11d), the maximum values of BP–CNTs and BP–CNTs/CNTs-based SCs without 4-NBD modification are 47.6 and 70.3 mWh cm$^{-3}$, which are lower than that of BP–CNTs (73.6 mWh cm$^{-3}$) and BP–CNTs/CNTs (96.5 mWh cm$^{-3}$), as shown in Fig. 3j. It is revealed that 4-NBD modification can improve the stability of BP, resulting in enhancing the energy density of SCs.

**Flexibility and application of supercapacitors.** We further investigate the flexibility and deformability of SCs by continuously operating devices in bending, rotating, twisting and folding motions. Figure 4a shows the flexible SC as operated with continual bending movements. Slight increases in capacitances are achieved, corresponding to 99.6% (265.9 F cm$^{-3}$), 98.5% (263 F cm$^{-3}$), 102.1% (272.3 F cm$^{-3}$) and 100.5% (268.3 F cm$^{-3}$) retentions of the initial flat-state capacitance (267 F cm$^{-3}$) under bending at 45°, 90°, 135° and 180° angles, respectively. Additionally, the capacitive retentions are found to be 97.6% (260.1 F cm$^{-3}$), 95.5% (255 F cm$^{-3}$), 98.5% (263 F cm$^{-3}$) and 97.4% (260 F cm$^{-3}$), after 1000 continuous cyclic tests in separated 45°, 90°, 135° and 180° bending angles, respectively. Furthermore, CV curves under twisted and rotated states show no notable changes relative to that of the flattened state (Fig. 4b), verifying the excellent deformability of the device. Although the performance is declined after folding, 90.1% of the specific capacitance is preserved, which is attributed to the favourable mechanical properties of the flexible SC (Young's modulus, 122 MPa; break elongation, 53.49%) (Supplementary Figure 12). Thus, these results demonstrate the SCs not only are highly flexible and foldable but also exhibit outstanding structural and electrochemical stability. To satisfy the improvement of energy storage performance for practical applications, we integrate three SCs in parallel (Supplementary Figures 13, 14). The output current of the assembled flexible SCs is correspondingly increased; the discharge time is three times longer than that of a single flexible SC.

Based on these remarkably favourable electrochemical performances, we explore flexible SCs as energy storage devices to power electronics for potential application. As shown in Fig. 4c, Supplementary Figures 15, 16 and Supplementary Movie 1,

flexible SCs with higher power output are used to light up 17 and 16 LEDs consisting of 'BP' and 'LED' shapes, respectively (3 V, 10 mA). Notably, there are no detectable changes in the brightness of the LEDs when undergoing 45°, 90° and 180° bending angles. Furthermore, a flexible SC is integrated on a textile to power a watch (3 V, 15 mA) (Fig. 4d, Supplementary Figure 17 and Supplementary Movie 2). To further improve the output energy of the SCs, we create a capacitive pouch with a high solid content of fabric electrodes. As illustrated in Fig. 4e, Supplementary Figure 18 and Supplementary Movie 3, a multi-colour display is successfully powered by integrated two capacitive pouches in series (3.8 V, 60 mA). The monochrome display is also brightly powered by a single capacitive pouch (Supplementary Figure 19 and Supplementary Movie 4). Therefore, our flexible SCs with high energy density, flexibility, deformability and foldability offer great potential to substantially replace flexible batteries.

**Mechanism of high-performance flexible supercapacitors.** On the basis of a comprehensive analysis of the electrode structure and electrochemical behaviour, the mechanism of high-performance flexible SC is illustrated in Fig. 5. For our flexible SC (Fig. 5a), the highest energy density (Fig. 5b) originates primarily from the designed BP–CNTs structure and MST fabrication, which is illustrated in the following aspects.

First, due to high-heat treatment, the stable hierarchical 2D/1D hetero-structure, where CNTs bridged within the BP effectively avoid restacking of nanosheets, exhibits an SSA increasing from 70.19 to 109.46 m$^2$ g$^{-1}$. Thereby, plentifully well-defined micro/mesopores (ionic-channels, particularly those measuring <1 nm) are formed within the network (see Fig. 2l). As shown in the detailed analysis in Supplementary Table 1, contributions of pore distribution to SSA are 131.6% of micropores and 22.1% of mesopores, revealing that micropores contribution is larger than that of mesopores. Similar to carbon-based materials, ion diffusion paths are shortened through the formation of those micro/mesopores, which facilitates faster ion transport and local accommodation[17,38]. We also hypothesize that the micro/mesopores smaller than 1 nm contribute much more to the capacitance because they are closer to the ion sizes[78]. In addition, the embedded CNTs enhance the interlayer electric pathways[40] so that the BP efficiently imparts its electrochemical activity. More importantly, this unique sheet/wire architecture can minimize the ion transmission distance and facilitate smooth ion motion at the electrode–electrolyte interface.

Second, we have developed MST for scalable fabrication of flexible electrodes. In particular, microfibres prepared by MST are assembled into non-woven fibre fabrics. As a result of fibres assembly, the nanoscale effects of the individual microfibre, including the unique porous structure, electrochemical activity, electrical conductivity and flexibility are amplified to apply to the whole non-woven fabric. Accordingly, the non-woven film-based flexible SC displays a high overall capacitance and energy density.

The refined mechanism is illustrated in Fig. 5c, d. When SCs are charged, fewer ions are accommodated in the pure CNTs network because wire-conduction-based EDLC is limited to smaller energy storage. Similar to graphene-based 2D structural materials (e.g. reversible ion absorption on an electrode–electrolyte interface[57,58]), the unique BP addition with appropriate interlayer distance[53], porous network and electrochemical activity[48] can cause more ions accumulation in the CNTs/BP electrodes[54]. However, poor interlayer conduction in BP limits the performance. By manipulating the in situ BP bridged with CNTs, better interlayer conduction, alleviated restacking and developed porosity leads to an opened 2D structure, pathway channels and redox process, resulting in ion-

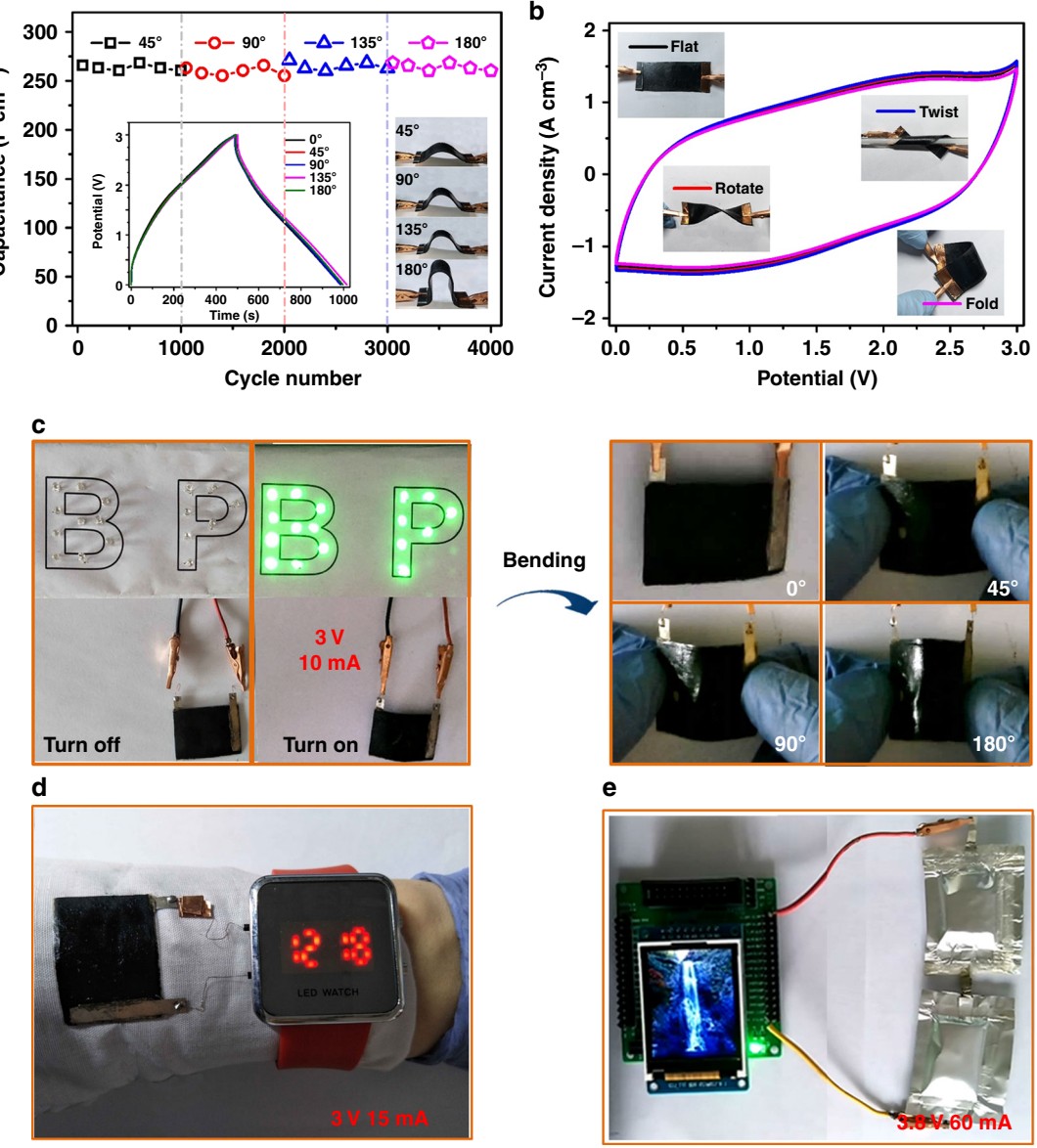

**Fig. 4** Flexibility, stability and application of supercapacitors. **a** Bending cyclic stability of a physical mixture of carbon nanotubes and hybrid carbon nanotubes chemical-bridged black phosphorous (CNTs/BP–CNTs) based supercapacitor (SC) at a current density of 0.4 A cm$^{-3}$. Insets: galvanostatic charge/discharge curves and photographs under different bending angles. **b** Cyclic voltammetry (CV) curves of SCs under flatted, twisted, rotated and folded states at a scan rate of 10 mV s$^{-1}$. **c** Photographs of supercapacitor stably lighting up light-emitting-diodes (LEDs) under different bending angles. **d** Photographs of an SC powering watch. **e** Photographs of two capacitive pouches integrated in series to power a multi-colour display. Guan Wu is the creator of the waterfall photo in the powered electronic device

flooding accommodation in the CNTs/BP–CNTs network. Therefore, benefiting from the composite nanomaterial, 2D/1D structural design and MST fabrication, flexible SC displays a higher energy storage behaviour.

## Discussion
In conclusion, we demonstrate chemically bridged hetero-structured BP–CNTs, and we propose an MST strategy to continuously fabricate high-performance flexible SCs based on non-woven fibre fabric electrodes. The as-constructed SCs are flexible enough to withstand bending, rotating, twisting and folding deformation. The designed BP–CNTs hybrid with an open layered structure, lamellar electron conduction and mechanical stability exhibits ionic pathways and redox activity for ion flooding accommodation. As a result, our solid-state flexible SCs exhibit a large specific volumetric capacitance of 308.7 F cm$^{-3}$,

high energy density of 96.5 mW h cm$^{-3}$, excellent life cycle stability (90.2% retention of the initial capacitance after 10000 cycles) and long-term bending durability. Furthermore, MST can boost the nanoscale effects of individual microfibres, such as their unique hetero-structure, electrochemical activity, electrical conductivity, flexibility and deformability. As a result, our flexible SCs can successfully power various electronics, including LEDs, smart watches and displays. Considering these outstanding achievements, we believe our available MST fabrication strategy will guide the new architecture for designing multifunctional composite electrodes and advance the progress of next-generation energy storage devices.

## Methods
**Synthesis of black phosphorous–carbon nanotubes.** The CNTs-doped BP crystals were synthesized by the mineralizer-assisted gas-phase transformation

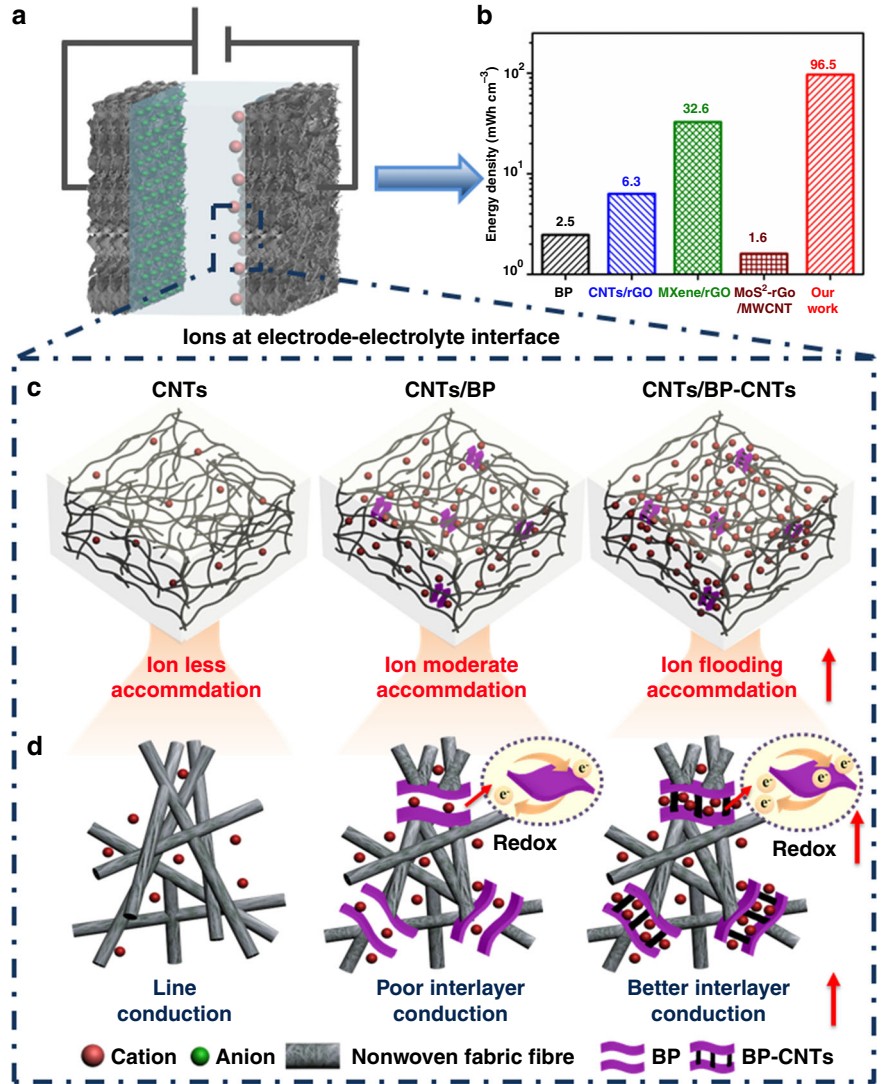

**Fig. 5** Mechanism for flexible supercapacitors. **a** Schematic illustration of flexible supercapacitor. **b** Energy density of our supercapacitor compared with that of other electrode-based supercapacitors. **c** Charge distribution in three kinds of designed electrodes. **d** Fine distribution of ions in three kinds of designed electrodes. Carbon nanotubes is abbreviated as CNTs. BP is defined as black phosphorous. CNTs/BP refers to the physical mixture of carbon nanotubes and black phosphorous, CNTs/BP–CNTs refers to the physical mixture of carbon nanotubes and carbon nanotubes chemical-bridged black phosphorous

method[51], which was reported previously[1]. Red phosphorus (500 mg, Chempur 99.999%), Sn (20 mg, Chempur 99.999%), $SnI_4$ (10 mg, Chempur 99.999%) and CNT (5 mg) were sealed in evacuated silica glass tubes (length: 180 mm; inner diameter: 8 mm) as the growth precursor. The tubes were placed horizontally in the heating zone of a quartz tube furnace. First, the furnace was heated to 750 °C within 3 h and held at 750 °C for 1 h. At this stage, the precursor evaporated, and red phosphorus gas reacted with Sn and $SnI_4$ gas to form $Sn_{24}P_{19.3}I_8$ compound, which acts as a nucleation agent for BP growth. Then, the furnace was cooled to 500 °C within 7.5 h and held for another 3 h at this temperature. During this growth stage, the gas-condensed, red phosphorous reacted with $Sn_{24}P_{19.3}I_8$ and transformed to BP[2]. With further growth, increasingly more red P transferred to BP as CNTs filled the gaps of layered BP to form the BP–CNT compound. The furnace was then cooled to 150 °C over 8 h before a final cooling step to room temperature. The CNT-doped BP crystals were thereby grown in the tube, and the BP was synthesized by this process.

**Synthesis of modified black phosphorous–carbon nanotubes**. The bulk of BP–CNTs (100 mg) were dispersed in acetonitrile (100 mL) under 200 W horn sonication treatment for 60 min (2 s on and 5 s off in an ice water bath). Then, 4-nitrobenzene-diazonium (4-NBD, 240 mg) and tetrabutylammonium hexa-fluorophosphate (3.87 g) were added to the stripped BP–CNT dispersion, which was maintained for 30 min. The mixture was then filtered and washed with acet-onitrile several times. The BP was modified during this process.

**Fabrication of the microfluidic spinning device**. The microfluidic spinning device consisted of two parts: syringe pumps (Nanjing Janus New Materials Co., Ltd) and a triphase microfluidic chip. The microfluidic chip was fabricated with inexpensive and easily obtained materials including a glass capillary (Kate experimental equipment business city company), epoxy resin AB glue (Shanghai Shu Da chemical technology company), silicone tube and polytetrafluoroethylene (PTFE) tubes (Kate experimental equipment business city company). Briefly, as external phase 1 and 2, two silicone tubes (inner diameter: 600 μm) were vertically inserted on the inside of the PTFE tubes (inner diameter: 1 mm), the outer phase 1 was located 1 cm away from the top of the PTFE tubes, and the outer phase 2 was located in the middle of the PTFE tubes. Then, the internal phase glass capillary (inner diameter: 150 μm) was coaxially inserted into the PTFE tubes, to an insertion length of approximately 3 cm. We used the epoxy resin AB glue instantaneous adhesive for sealing and fastening it. Finally, we fixed the channel on the glass using the epoxy resin AB glue.

**Fabrication of non-woven fibre fabrics**. First, BP–CNTs modified by 4-NBD were sonicated in N,N-dimethylformamide (DMF), forming a uniform dispersion. Then, a mixture of BP–CNT dispersion, CNTs and TPU was further ultrasonically processed and stirred in DMA for 24 h at 60 °C, forming an even dispersion. A triphase microfluidic microreactor with three microchannels composed of one core flow (well dispersion of BP–CNTs, CNTs and TPU in DMA), and two sheath flow ($C_2H_5OH$ and $H_2O$) was then incorporated. Typically, the core flow of the dis-persion (flow rate: 70 mL h$^{-1}$) was first injected into the microreactor by syringe,

followed by pumping the sheath flow of $C_2H_5OH$ (external phase 1, flow rate: 55 mL h$^{-1}$) into the device for pre-coagulation of dispersion. Subsequently, the dispersion was deep-coagulated to form microfibres by extruding water (external phase 2, flow rate: 40 mL h$^{-1}$) through solvent exchange. The as-prepared fibres were collected by filtration and then hot pressed and dried in an oven at 80 °C for 2 days, forming non-woven fibre fabrics. During the process of hot pressing, non-woven fibre fusion was realized through heating or simple exposure to solvent vapours. The solvent–vapour-mediation allows multiple junctions of fibres, which are interfused and interconnected with each other, forming non-woven fibre fabrics[62,66].

**Construction and characterization of a flexible supercapacitor**. First, the electrolyte layer was prepared as follows: a mixture of 2 g EMIBF$_4$ and 1 g PVDF-HFP was added to 20 mL DMF solution, which was heated at 80 °C under vigorous stirring to obtain the uniform solution. The 3 mL solution was cast onto a glass substrate and dried to obtain an electrolyte layer. Then, a half-dried electrolyte layer sandwiched between two compressed fabric electrodes was used to construct a flexible SC at 60 °C for 1 day. For the SCs, the CV, galvanostatic cycling and EIS were conducted using a CHI760E electrochemical workstation. The specific volumetric capacitance of SCs based on galvanostatic cycle test was calculated by $C_v = \frac{4I\Delta t}{V\Delta V}$, where $I$ (A), $\Delta t$ (s), $\Delta V$ (V), and $V$ (cm$^3$) are the discharge current, discharge time, voltage range of discharge during the discharge process, and total volume of the two fabric electrodes, respectively. The energy density and average power energy were calculated by $E = C_v V^2/8$ and $P = E/\Delta t$, respectively, where $C_v$, $V$, and $\Delta t$ (s) are the volumetric capacitance, operating voltage and discharge time, respectively.

**Fabrication of the capacitive pouch**. First, five fabric films were hot-pressed into one high solid content fabric electrode. Then, an aluminum was connected to the fabric electrode (35 mm × 20 mm) as the positive electrode. A nickel terminal was connected to the other fabric electrode (35 mm × 20 mm) as a negative electrode. Subsequently, the prepared EMIBF$_4$/PVDF-HFP organic electrolyte was used as a separator film to allow ion transport between the electrodes while preventing a short circuit. In addition, the assembly was pre-heated. Finally, a capacitive pouch was obtained through encapsulating the SC filled with ionic liquids in an aluminum foil composite bag.

## Data availability
The data supporting the findings of this study are available from the corresponding authors upon reasonable request.

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

## Acknowledgements

This work was supported by the National Natural Science Foundation of China (21736006, 21706120, 21474052), Natural Science Foundation of Jiangsu Province (BK20170973), National Key Research and Development Program of China (2016YFB0401700), China Postdoctoral Science Foundation (2018M630549), Priority Academic Program Development of Jiangsu Higher Education Institutions (PAPD) and Fund of State Key Laboratory of Materials-Oriented Chemical Engineering (ZK201720, ZK201704).

## Author contributions

S.C. and G.W. planned and designed the project. X.W., G.W. and Y.X. conducted all of the experiments. H.C. helped to synthesize the materials. Q.Y. and Y.H. helped characterize the samples. K.Z. and W.C. helped to elaborate the mechanism. G.W., X.W. and S.C. analysed the data and wrote the paper. X.W., Y.X. and Y.H. contributed equally to this work. All authors reviewed the manuscript.

## Additional information

**Competing interests:** The authors declare no competing interests.

