## [Peer Review File · Nature Communications]

Reviewers' comments:

Reviewer #1 (Remarks to the Author):

The manuscript reports a hetero-structured material made of black phosphorous (BP) chemical-bridged with carbon nanotubes (CNTs), which can be further constructed into non-woven fiber fabrics via microfluidic-spinning-technique (MST) and used as SC electrodes. The flexible SC exhibits remarkably high energy-density (96.5 mWh cm^{-3}), large volumetric capacitance (308.7 F cm^{-3}), long life cycle

stability and excellent deformable durability. Due to the good performance, it can power the electronics and demonstrate a promise for the practical application. The work shows a novel design and fabrication of the flexible SC. There are still some questions that need to be addressed before consideration in Nature Communications.

1. In Figure 2j-k, C 1s XPS spectra can demonstrate the P-C formation between BP and 4-NBD. But it is weak to claim the presence of P-C bonds in BP-CNTs. The XPS spectra of BP-CNTs should be provided to show the presence of P-C bonds in this composite.

2. The authors should provide the information of TPU. The readers do not know what TPU stands for.

3. 4-NBD was used to stabilize BP in this work. However, 4-NBD can also react with CNTs (J. Am. Chem. Soc., 2003, 125, 1156–1157). Do the authors consider this influence?

4. Does 4-NBD contribute to improving the energy density of BP? The authors should provide the cycling stability and energy density of 4-NBD modified BP.

Reviewer #2 (Remarks to the Author):

The manuscript entitled "UMicrofluidic-Spinning-Constructed Non-Woven Fabrics of Hetero-Structured Black Phosphorous-Based Hybrid Microfibers toward High Energy-Density Flexible Supercapacitor" presents non-woven fiber fabrics via microfluidic-spinning-technique of hybrid black phosphorous-carbon nanotubes as wearable supercapacitor electrode materials. The flexible supercapacitor exhibits remarkably high energy-density, large volumetric capacitance, long life cycle stability and excellent deformable durability. In general, this work is interesting. However, some problems were raised after reviewing this manuscript, as follows:

1. Provide the Raman results for hybrid black phosphorous-carbon nanotubes.

2. Some related important references should be cited/mentioned.

3. In Fig 3(j), more important comparative data should be added into the figure.

4. Please provide the phase angle vs frequency.

5. Please provide the RC time constant.

Reviewer #3 (Remarks to the Author):

The article by Wu et al. Describes the creation of supercapacitors. The here used hybrid materials, black phosphorous bridged on carbon nanotubes (BP-CNTs) seems to be advantageous for several reasons; (i) the high capacity /energy storage ability, (ii) the ease of implementing these particles into a fibre-like material that is flexible and (iii) enhanced electron shuttling capability. In addition to the synthesis and characterization of the material, a mechanism of electron storage and applications for operating LEDs and displays are presented in addition.

The paper has some interesting aspects for finding new and flexible supercapacitors, however, a large fraction of the study appears to be a modified version of the work by Yang et al, which appeared in ACS Appl. Materials and Interfaces 9 (2017), p. 44478. The paper does not appear in the reference

list, and my recommendation is that the authors have a look on this study, where sheets of BP-CNT are created and characterized and reorganize their manuscript accordingly.

Although I think Wu et al. present data beyond this before mentioned paper and improved the BP-CNT supercapacitor, I still believe that the study is not of sufficient novelty any more for publication in Nature Communication. The added aspects are the use of microfluidic spinning technology (where the benefit is questionable), and the presentation of the "real" use (which I like, but it is not quite spectacular).

A few minor concerns on the all-over manuscript and the aspect of microfluidics:

Language:

- The title includes many keywords, but is difficult to understand after first reading
- There are many sentences that are unclear or have grammar mistakes, e.g lines 224-225 (remove was), or lines 226-227 (compress, change to compressed), line 373 (resatcking change to restacking?) or lines 281-284, to mention a few of them

Figures:

- High-content figures, but this is for the sake of clearness. In addition, the figure captions need revision, e.g. Figure 1 depicts a lot of information, but the caption is extremely short.

Microfluidic spinning:

- This hydrodynamic focusing unit is not well-described. There is a photograph in the Supplementary information (S4b), which indicates that glass was used together with glue and tubings. No indication, how the device was made, how it was assembled, what dimensions where used, or how the product was taken out. Indeed, figure 3a shows a device that must have several inlets (Bp-CNTs/ethanol/water), but this does not match figure 4b in the supplementary information. Figure 3a is supposed to show the 80 μm fibres (line 225), but this is not shown at all.
- Water is added after ethanol, but there is nothing mentioned, where the ethanol remains. The authors state that water is extruded (l. 222) and the solvent is exchanged, but neither the extrusion nor the exchange of solvent is shown or proven, and I hardly believe that the ethanol "disappears". Is it maybe diluted?
- The step from the microfluidic focusing to the formation of the macroscopic fabric with cm-dimensions is not clear (fig.3a and Fig. S4c), as well as the reason to produce fabrics of different forms (Fig 3a right image).
- The throughput seems to be low..?

The response to reviewers' comments

Reviewer #1:

The manuscript reports a hetero-structured material made of black phosphorous (BP) chemical-bridged with carbon nanotubes (CNTs), which can be further constructed into non-woven fiber fabrics via microfluidic-spinning-technique (MST) and used as SC electrodes. The flexible SC exhibits remarkably high energy-density (96.5 mWh cm^{-3}), large volumetric capacitance (308.7 F cm^{-3}), long life cycle stability and excellent deformable durability. Due to the good performance, it can power the electronics and demonstrate a promise for the practical application. The work shows a novel design and fabrication of the flexible SC. There are still some questions that need to be addressed before consideration in Nature Communications.

1. In Figure 2j-k, C 1s XPS spectra can demonstrate the P-C formation between BP and 4-NBD. But it is weak to claim the presence of P-C bonds in BP-CNTs. The XPS spectra of BP-CNTs should be provided to show the presence of P-C bonds in this composite.

Reply: Thanks very much for your valuable suggestions. According to your opinion, we have measured XPS of BP and BP-CNTs. As clearly shown in Figure a and b, the BP-CNTs shows P-C formation at $\sim 284 \text{ eV}$ binding energy in the C1s spectra, which evidently conform the presence of P-C bonds in chemically bridged BP-CNTs.

Figure. High resolution C1s XPS spectra of BP (a) and BP-CNTs (b).

The related revision is on page 9:

“X-ray photoelectron spectroscopy (XPS) is carried out to clarify the typical C1s and P2p peaks, characterizing the formation of P-C bonds in BP-CNTs^{49,54,62}. Obviously, the BP-CNTs shows P-C formation at ~284 eV binding energy in the C1s spectra in Fig. 2j and 2k, which evidently conforms the chemical bridge between BP and CNTs under high-heat treatment.”

Figure 2 | Structural characterization of BP-CNTs. (a), and (b) TEM images of BP-CNTs at low and high magnifications, respectively. (c) HRTEM image of BP-CNTs. (d) High magnification of the selected domain in (a). The corresponding EDS elemental mapping images of C (e) and P (f). Surface SEM images of BP-CNTs at low (g) and high magnifications (h), respectively. (i) Structural illustration of BP-CNTs. High resolution C1s XPS spectra of BP (j) and BP-CNTs (k). (l) The

micro-pore size distributions, insets are typical nitrogen adsorption and desorption isotherms, and meso-pore size distributions.

2. The authors should provide the information of TPU. The readers do not know what TPU stands for.

Reply: Thanks very much for your good suggestions. Based on your advice, we have provided the detailed information of TPU. TPU stands for thermoplastic polyurethane.

The related revision is on page 8:

“TPU is thermoplastic polyurethane”

3. 4-NBD was used to stabilize BP in this work. However, 4-NBD can also react with CNTs (J. Am. Chem. Soc., 2003, 125, 1156–1157). Do the authors consider this influence?

Reply: Thanks very much for your nice suggestions. Based on your suggestion, we have done controlled experiment to demonstrate whether the 4-NBD can react with CNTs or not. We use Raman spectra for this demonstration. As shown the Raman spectra in Figure A, CNTs treated with 4-NBD shows no changes in the Raman peaks when compare to pure CNTs. It is indicated that there is no chemical reaction between 4-NBD and CNTs in our system. As the reviewer mentioned, the reaction between 4-NBD and CNTs requires special conditions. As described in the literature (J. Am. Chem. Soc., 2003, 125, 1156-1157), the addition of isoamyl nitrite and heating (60°C) are the necessary condition for their

chemical reacting. However, in our system, 4-NBD and CNTs are mixed in acetonitrile at room temperature for 30 minutes and there is no chemical reaction.

Figure A Raman spectra of CNTs and CNTs treated with 4-NBD.

4. Does 4-NBD contribute to improving the energy density of BP? The authors should provide the cycling stability and energy density of 4-NBD modified BP.

Reply: Thanks very much for your valuable suggestions. In our work, because the BP and BP-CNTs are unstable in air condition, the electrochemical performance of BP and BP-CNTs based hybrid electrodes are all 4-NBD modified. Thus, to investigate whether 4-NBD contribute to improving the energy density of BP, we have done the experiment of BP and BP-CNTs without 4-NBD modification. As shown in Figure a, b and c, BP/CNTs and BP-CNTs/CNTs based SCs with 4-NBD modification display longer discharge time than that of electrodes without

4-NBD modification. It is due to the oxidation of BP that decreases the electrochemical performance. Figure d shows the specific volumetric capacitances of SCs. Clearly, BP/CNTs and BP-CNTs/CNTs based SCs without 4-NBD modification present specific capacitance of 152.3 and 225.1 F cm⁻³ at 0.1 A cm⁻³, which are smaller than that of 4-NBD modification BP/CNTs (235.5 F cm⁻³) and BP-CNTs/CNTs (308.7 F cm⁻³) in Fig. 3g. Furthermore, the cycle stabilities of BP/CNTs and BP-CNTs/CNTs without 4-NBD modification are declined obviously when compare with BP/CNTs and BP-CNTs/CNTs with 4-NBD modification (Figure e). It is indicated that 4-NBD modification can improve the stability of SCs. Figure f is energy and power densities of SCs. BP/CNTs and BP-CNTs/CNTs based SCs without 4-NBD modification perform the highest energy densities of 47.6 and 70.3 mWh cm⁻³, which is lower than that of BP/CNTs (73.6 mWh cm⁻³) and BP-CNTs/CNTs (96.5 mWh cm⁻³) in Fig. 3j. It is revealed that 4-NBD modification can improve the stability of BP, which improve the energy densities of SCs.

Figure | (a) Galvanostatic charge/discharge curves of 4-NBD modified BP/CNTs and without modified BP/CNTs at the current density of 0.1 A cm^{-3} . (b) Galvanostatic charge/discharge curves of 4-NBD modified BP-CNTs/CNTs and without modified BP-CNTs/CNTs at the current density of 0.1 A cm^{-3} . (c) Galvanostatic charge/discharge curves of CNTs/BP and CNTs/BP-CNTs without 4-NBD modification at the current density of 0.1 A cm^{-3} . (d) The calculated specific capacitances under different current densities. (e) Cycle stability of SCs under a voltage of 3

V at a current density of 0.4 A cm^{-3} . (f) Energy density versus power density of SCs.

The related revision is on page 17:

“In order to investigate the effect of 4-NBD, the performances of CNTs/BP and CNTs/BP-CNTs without modification are applied to comparison. As illustrated in Supplementary Fig. 11a and Fig. 3f, CNTs/BP and CNTs/BP-CNTs without 4-NBD modification display worse charge storage performance than that of electrodes with 4-NBD modification. It is mainly due to the oxidation of BP that decreases the electrochemical performance. Clearly, BP/CNTs and BP-CNTs/CNTs based SCs without 4-NBD modification present specific capacitance of 152.3 and 225.1 F cm^{-3} at 0.1 A cm^{-3} (Supplementary Fig. 11b), which are smaller than that of 4-NBD modification BP/CNTs (235.5 F cm^{-3}) and BP-CNTs/CNTs (308.7 F cm^{-3}) in Fig. 3g. Furthermore, the cycle stabilities of BP/CNTs and BP-CNTs/CNTs without 4-NBD modification (Supplementary Fig. 11c) are declined more obviously when compare with BP/CNTs and BP-CNTs/CNTs with 4-NBD modification (Fig. 3i). It is indicated that 4-NBD modification can improve the stability of SCs. In construct to energy density, (Supplementary Fig. 11d), BP/CNTs and BP-CNTs/CNTs based SCs without 4-NBD modification show the highest values of 47.6 mWh cm^{-3} and 70.3 mWh cm^{-3} , which are still lower than that of BP/CNTs (73.6 mWh cm^{-3}) and BP-CNTs/CNTs (96.5

mWh cm⁻³) in Fig. 3j. It is revealed that 4-NBD modification can improve the stability of BP, resulting in enhancing the energy densities of SCs.”

Supplementary Fig. 11 | (a) Galvanostatic charge/discharge curves of CNTs/BP and CNTs/BP-CNTs without 4-NBD modification at the current density of 0.1 A cm⁻³. (b) The calculated specific capacitances under different current densities. (c) Cycle stability of SCs under a voltage of 3 V at a current density of 0.4 A cm⁻³. (d) Energy density versus power density of SCs.

Reviewer #2:

The manuscript entitled "Microfluidic-Spinning-Constructed Non-Woven Fabrics of Hetero-Structured Black Phosphorous-Based Hybrid Microfibers toward High Energy-Density Flexible Supercapacitor" presents non-woven fiber fabrics via microfluidic-spinning-technique of hybrid black phosphorous-carbon nanotubes as wearable supercapacitor electrode materials. The flexible supercapacitor exhibits remarkably high energy-density, large volumetric capacitance, long life cycle stability and excellent deformable durability. In general, this work is interesting. However, some problems were raised after reviewing this manuscript, as follows:

1. Provide the Raman results for hybrid black phosphorous-carbon nanotubes.

Reply: Thanks very much for your valuable suggestions. According to your opinion, we have provided the Raman results for hybrid black phosphorous-carbon nanotubes. As shown in Supplementary Fig. 4, the CNTs/BP mixture shows the Raman active modes of both CNTs and BP. The peaks at 362, 438 and 466 cm^{-1} correspond to A_{1g} , B_{2g} and A_{2g} modes of BP, respectively (ACS applied materials & interfaces, 2017, 9, 36849-36856, Angew. Chem. Int. Ed, 2018, 57, 4543-4548). Meanwhile, two strong peaks located at 1352 and 1591 cm^{-1} are signed to D and G bands of CNTs, respectively. It is indicated that the mixture of BP and

CNTs make no P-C bond. However, when taking chemical-bridged with CNTs for consideration, it is worth noting that a small peak at 703 cm^{-1} (Nano Lett 14, 2014, 4573-4580) in CNTs/BP-CNTs corresponds to P-C bond, revealing that a small portion of P-C bonds are formed in BP-CNTs.

Supplementary Fig. 4 | Raman spectra of CNTs/BP-CNTs and CNTs/BP.

The related revision is on page 10:

“As described in the Raman spectra (Supplementary Fig. 4), CNTs/BP mixture shows three peaks at 362 cm^{-1} , 438 cm^{-1} and 466 cm^{-1} , corresponding to A_{1g} , B_{2g} and A_{2g} modes of BP^{49,62}, respectively. Meanwhile, two strong peaks located at 1352 cm^{-1} and 1591 cm^{-1} are signed to D and G bands of CNTs, respectively. However, it is worth noting that a small peak at 703 cm^{-1} in CNTs/BP-CNTs attributes to P-C bond⁵⁴, revealing that a small portion of P-C bonds are formed in BP-CNTs.”

Supplementary Fig. 4 | Raman spectra of CNTs/BP-CNTs and CNTs/BP.

2. Some related important references should be cited/mentioned.

Reply: Thanks very much for your excellent suggestions. Based on your advice, we have cited some related important references. To demonstrate the structure of BP-CNTs, we have cited literatures (Angew. Chem. Int. Ed. 2018, 57, 4543-4548, Nano letter, 2014, 14, 4573-4580) for its Raman spectra. To demonstrate the capacitance level of our SCs reaching one of the highest values, we have cited literatures (carbon nanosheet is $\sim 100 \text{ F cm}^{-3}$ in Adv. Mater. 2018, 30, 1706054; $\text{MnO}_2/\text{carbon cloth}$ is 14.41 F cm^{-3} in ACS Nano 2018, 12, 3557-3567). To demonstrate the energy density level of our SCs reaching one of the highest values, we have cited literatures (graphene/PANI is 8.80 mWh cm^{-3} in Adv. Mater. 2018, 30, 1706054; carbon nanosheet is 8.4 mWh cm^{-3} in Adv. Mater. 2018, 30, 1706054; $\text{Co}_3\text{O}_4/\text{Co(OH)}_2$ is 9.4 mWh cm^{-3}

in Nano Energy, 2017, 35, 138-145).

3. In Fig 3(j), more important comparative data should be added into the figure.

Reply: Thanks very much for your nice suggestions. Based on your opinion, we have added more important comparative data. We have added literatures (carbon nanosheet is 8.4 mWh cm⁻³ in Adv. Mater. 2018, 30, 1706054; graphene/PANI is 8.80 mWh cm⁻³ in Adv. Mater. 2018, 30, 1706054; Co₃O₄/Co(OH)₂ is 9.4 mWh cm⁻³ in Nano Energy, 2017, 35, 138-145; MnO₂/carbon cloth is 8.3 mWh cm⁻³ in ACS Nano 2018, 12, 3557-3567). To the best of our knowledge, the maximum volumetric energy density of our SC is one of the best values among commercial product and relevantly composited 2D layered materials (**Supplementary Table 3**), such as Li thin-film battery, CNTs (2 mWh cm⁻³), rGO (2.5 mWh cm⁻³), BP (2.5 mWh cm⁻³), CNTs/rGO (6.3 mWh cm⁻³), **MnO₂/carbon cloth (8.3 mWh cm⁻³), carbon nanosheet (8.4 mWh cm⁻³), graphene/PANI (8.80 mWh cm⁻³), Co₃O₄/Co(OH)₂ (9.4 mWh cm⁻³), Ni@MnO₂ (11.1 mWh cm⁻³) and MXene/Graphene (32.6 mWh cm⁻³)**

Fig. 3j. Energy density versus power density of SCs compared with other electrodes based energy-storage systems.

Supplementary Table 3 | Energy density values of our SC compared with previously reported flexible SCs.

	Electrode materials	Energy density (mWh cm ⁻³)	Reference
1	MoS ₂ -rGO/MWCNTs	1.6	1
2	CNTs	2	2
3	rGO	2.5	3
4	Black Phosphorus	2.5	4
5	CNTs/rGO	6.3	5
6	MnO ₂ /carbon cloth	8.3	6
7	carbon nanosheet	8.4	7
8	graphene/PANI	8.8	8
9	Co ₃ O ₄ /Co(OH) ₂	9.4	9
10	Ni@MnO ₂	11.1	10
11	MXene/Graphene	32.6	11
12	Li thin-film battery	4V/500 μAh	12
13	Our work	96.5	

4. Please provide the phase angle vs frequency.

Reply: Thanks very much for your kind suggestions. According to your advice, we provide the phase angle vs frequency. **The dependence of phase angle on the frequency for SCs is shown in Supplementary Fig 10.** For CNTs/BP-CNTs SC, the phase angle is close to -90° at low frequencies, indicating the ideal capacitive behavior. The characteristic frequency f_0 at a phase angle of -45° or its corresponding relaxation time, also called RC time constant ($\tau_0=1/f_0$) marks the point where resistive and capacitive impedances are equal (Nat Commun, 2014, 5, 4554; Advanced Functional Materials, 2012, 22, 4501-4510). The CNTs/BP-CNTs, CNTs/BP and CNTs exhibit an f_0 of 0.202 Hz, 0.14 Hz and 0.107 Hz, respectively, which corresponded to a RC time constant of 4.95 s, 7.14 s and 9.35 s, respectively. It is implied that the faster frequency response of CNTs/BP-CNTs shows enhanced ion transport rate within the hetero-structured electrodes.

Supplementary Fig. 10 | Bode plots of Phase angle vs frequency curves.

The related revision is on page 16:

“Supplementary Fig. 10 shows the dependence of phase angle on the frequency for SCs. For CNTs/BP-CNTs SC, the phase angle is close to -90° at low frequencies, indicating the better ideal capacitive behavior than that of CNTs/BP and pure CNTs.”

5. Please provide the RC time constant.

Reply: Thanks very much for your excellent suggestions. Based on your advice, we provide the RC time constant. The characteristic frequency f_0 at a phase angle of -45° or its corresponding relaxation time, also called **RC time constant** ($\tau_0 = 1/f_0$) marks the point where resistive and capacitive impedances are equal (Nat Commun, 2014, 5, 4554; Advanced Functional Materials, 2012, 22, 4501-4510). The CNTs/BP-CNTs, CNTs/BP and CNTs exhibit an f_0 of 0.202 Hz, 0.14 Hz and 0.107 Hz, respectively, which corresponded to a RC time constant of

4.95 s, 7.14 s and 9.35 s, respectively. It is implied that the faster frequency response of CNTs/BP-CNTs shows enhanced ion transport rate within the hetero-structured electrodes.

Supplementary Fig. 10 | Bode plots of Phase angle vs frequency curves.

The related revision is on page 16:

“The characteristic frequency f_0 at a phase angle of -45° or its corresponding relaxation time, also called RC time constant ($\tau_0=1/f_0$) marks the point where resistive and capacitive impedances are equal^{6,72}. The CNTs/BP-CNTs, CNTs/BP and CNTs exhibit an f_0 of 0.202 Hz, 0.14 Hz and 0.107 Hz, respectively, which corresponds to a RC time constant of 4.95 s, 7.14 s and 9.35 s, respectively. It is implied that the faster frequency response of CNTs/BP-CNTs shows enhanced ion transport rate within the hetero-structured electrodes.”

Reviewer #3:

The article by Wu et al. describes the creation of supercapacitors. The here used hybrid materials, black phosphorous bridged on carbon nanotubes (BP-CNTs) seems to be advantageous for several reasons; (i) the high capacity /energy storage ability, (ii) the ease of implementing these particles into a fibre-like material that is flexible and (iii) enhanced electron shuttling capability. In addition to the synthesis and characterization of the material, a mechanism of electron storage and applications for operating LEDs and displays are presented in addition.

1. The paper has some interesting aspects for finding new and flexible supercapacitors, however, a large fraction of the study appears to be a modified version of the work by Yang et al, which appeared in ACS Appl. Materials and Interfaces 9 (2017), p. 44478. The paper does not appear in the reference list, and my recommendation is that the authors have a look on this study, where sheets of BP-CNT are created and characterized and reorganize their manuscript accordingly.

Reply: Thanks for your good suggestions. According to your opinions, we have a carefully look on your recommended literature (ACS Appl. Mater. Interfaces 2017, 9, 44478-44484). Although this work has made the progress, our work is greatly innovative instead of the modified version of Yang et al work. **Novelty and innovation is as follows.**

1) In our work, we designed hetero-structured BP-CNTs materials, in

which the CNTs within BP is in-situ chemically bridged through P-C bonds. Thus, this chemical-bridged design can enhance the electrical conductivity, mechanical stability and restacking of BP. Additionally, the architected BP-CNTs hybrid materials also create more ionic channels (micro/meso-pores, especially $< 1\text{nm}$) for ions fast diffusion and flooding storage. **From the materials design, our hetero-structured BP-CNTs material is innovative.** Our flexible SCs display remarkably high energy density (96.5 mWh cm^{-3}), and large specific volumetric capacitance (308.7 F cm^{-3}). However, in Yang et al work, BP was just mixed with CNTs through physical sonication. In this case, the poor interfacial contact between stacked BP and CNTs resulted in the bad electron conduction and stability, which decreased the capacitance and life stability. Thus, they obtained a relatively low capacitance (35.7 F cm^{-3}) and energy density (5.71 mWh cm^{-3}).

2) In our work, we first proposed the microfluidic-spinning-technique (MST) to construct BP-CNTs based microfibers and fibers assembled non-woven fabric. **From the fabrication method, our MST is innovation in the preparation of flexible electrodes.** It is the utilization of MST that allows fabric electrodes with outstanding mechanical strength (Young's modulus, 313 MPa ; break elongation, 17.96%), which ensure the flexibility and deformability of

supercapacitors (SCs). Thus, the as-constructed SCs are flexible enough to withstand bending, rotating, twisting and folding deformation. Also, the mechanical properties (especially of break elongation) exceed the values reported in most literatures. Our photos and videos can prove the advantages of MST method. However, in Yang et al work, there is no solid evidence to demonstrate the benefits of their methods.

3) In this work, we realize the practical applications of SCs to effectively power a variety of electronics. As we all known, in our field, SCs are basically limited to power LEDs and watch with low start-up current. However, we have successfully realized the SCs to power not only 17 LEDs with bending deformation but also monochrome and multi-color displays with higher start-up current, which is nearly close to the current of the battery. **From the practical applications, our SCs is innovation to power potential electronics with different current ranges.** However, in Yang et al work, only 1 LED could be lighted up, not to mention displays. Additionally, there is no video to support their application. Thus, in future, we want to realize the flexible SC to promising replace batteries to power mobile phone.

4) In our work, we used EMIBF₄/PVDF-HFP as solid-state electrolyte layer, which could be applied in wider potential window (0~3V). We

do not need to manage complex ways to achieve higher voltage and current output.

Therefore, considering of the materials design, fabrication method and practical applications, our work is of great novelty and innovation instead of the modified version of Yang et al work. Also, thanks for your suggestion, we have cited Yang et al work in our manuscript.

2. Although I think Wu et al. present data beyond this before mentioned paper and improved the BP-CNT supercapacitor, I still believe that the study is not of sufficient novelty any more for publication in Nature Communication. The added aspects are the use of microfluidic spinning technology (where the benefit is questionable), and the presentation of the "real" use (which I like, but it is not quite spectacular).

Reply: Thanks for your suggestions. Microfluidic spinning technique is a power platform, which is widely used in not only academic science but also industrial community. Microfluidic systems have aroused enormous concern because of its several advantages including organized morphologies and compositions, facile manipulation, continuous production, easy-controllability, benign reaction conditions and environmental protection.

In academic research, based on the microfluidic systems, the microfibers with identical performance can be in situ synthesized in a microfiber reactor through the microfluidic spinning technique without

other complex conditions (Nature materials, 2011, 10:877-883. Nature chemistry, 2009, 1:165-165). In the potentially industrial production, microfluidic systems can be used for large-scale preparation of polymer emulsion and micro-droplets, where the production rate of this system can reach to 277 g h^{-1} and $1 \text{ trillion h}^{-1}$ (Nature communication, 2018, 9:1222). Also, submillisecond organic synthesis that outpaces the very rapid anionic Fries rearrangement to chemoselectively functionalize iodophenyl carbamates was realized using microfluidic technique (Science, 2016, 352, 691-694) In addition, American pharmaceutical companies invest 60 million dollars in microfluidics systems, and the annual throughput of drugs can reach to thousands tones. Therefore, microfluidic system has been applied in related fields, such as pharmaceutical factory and coating factory.

In our work, although the production speed of fiber is still a bit low, we will do our best to explore and improve our skills to realize the real faster and large-scale production of fibers. In the future, we have to achieve the large-area woven fiber-based supercapacitor to power mobile phone.

3. A few minor concerns on the all-over manuscript and the aspect of microfluidics:

Language:

-The title includes many keywords, but is difficult to understand after first reading

Reply: Thank you for the nice suggestion. To make the title more understandable, we have changed the title into “Microfluidic-Spinning-Construction of Black Phosphorous Hybrid Microfiber-Based Non-Woven Fabrics toward High Energy-Density Flexible Supercapacitor”.

-There are many sentences that are unclear or have grammar mistakes, e.g lines 224-225 (remove was), or lines 226-227 (compress, change to compressed), line 373 (resatcking change to restacking?) or lines 281-284, to mention a few of them

Reply: Thank you for the kind suggestion. According to your opinions, we have checked the whole manuscript and revised the grammar mistakes in our paper. We have removed “was”. We have corrected mistakes “compress”, “into “compressed”. We have corrected mistakes “resatcking”, “into “restacking”. We have corrected mistakes “facilitate”, “into “facilitates”. We have corrected mistakes “is”, “into “are”. We have corrected mistakes “SCs”, “into “supercapacitors (SCs)”. We have corrected mistakes “application”, “into “applications”. We have corrected mistakes “material”, “into “materials”.

The related revision is as follows:

“(On page 2) We have corrected mistakes “SCs”, “into “supercapacitors (SCs)

(On page 2) We have corrected mistakes “application”, “into

“applications.

(On page 2) We have corrected mistakes “material”, “into “materials

(On page 13) We have removed “was”.

(On page 13) We have corrected mistakes “compress”, “into “compressed”.

(On page 23) We have corrected mistakes “resatcking”, “into “restacking”.

(On page 5) We have corrected mistakes “facilitate”, “into “facilitates”.

(On page 9) We have corrected mistakes “is”, “into “are”.

4. Figures:

-High-content figures, but this is for the sake of clearness. In addition, the figure captions need revision, e.g. Figure 1 depicts a lot of information, but the caption is extremely short.

Reply: Thanks very much for your valuable suggestions. Based on your opinion, we have written the caption in detail of Fig.1 and Fig. 3.

The related revision is as follows:

“(On page 8) **Figure 1 | Schematic illustrations.** (a) The synthesis of BP-CNTs under heat treatment BP and CNTs, and Chemical passivation of BP-CNTs using 4-NBD. (b) MST fabrication of microfibers via a triphase microfluidic device consisting of one core flow and two sheath flow and fibers assembled into no-woven fabrics, which can be cut into different shapes. TPU is thermoplastic polyurethane. (c) Construction of

flexible SC by hot-press two conductive fabric layers and one polymer-supported ionic liquid electrolyte layer. The SCs are of potential to power various electronics for application.

(On page 19) **Figure 3 | Electrochemical performance of flexible SCs.**

(a) MST fabrication of fibers-based no-woven fabrics. Left part is the illustration of microchannel of device. Middle part is the photo of no-woven fabric. The inset is the SEM image of fabric. Right part is the no-woven fabric that can be cut into various shapes and applied with different deformation. (b)~(d) Schematic illustration of the designed electrode structure. (e) CV curves of SCs at a scan rate of 10 mV s^{-1} . (f) Galvanostatic charge/discharge curves at the current density of 0.1 A cm^{-3} . (g) The calculated specific capacitances under different current densities. (h) EIS analysis of SCs. The inset figures are the depressed semicircle of Nyquist plots and the equivalent circuit model. Symbols denote experimental data, while the green lines represent the fitted data. (i) Cycle testing of SCs under a voltage of 3 V at a current density of 0.4 A cm^{-3} , the inset: Galvanostatic charge/discharge curves after 10,000 cycles. (j) Energy density versus power density of SCs compared with other electrodes based energy-storage systems. The used BP and BP-CNTs are all modified by 4-NBD.”

5. Microfluidic spinning:

-This hydrodynamic focusing unit is not well-described. There is a

photograph in the Supplementary information (S4b), which indicates that glass was used together with glue and tubings. No indication, how the device was made, how it was assembled, what dimensions where used, or how the product was taken out. Indeed, figure 3a shows a device that must have several inlets (Bp-CNTs/ethanol/water), but this does not match figure 4b in the supplementary information. Figure 3a is supposed to show the 80 μm fibres (line 225), but this is not shown at all.

Reply: Thank you for the valuable review. According to your review, we introduce the process of microfluidic spinning in detail.

Fabrication of microfluidic spinning device

The microfluidic spinning device consists of two parts: syringe pumps (Nanjing Janus New Materials Co., Ltd) and triphase microfluidic chip. The microfluidic chip was fabricated with cheap and easily obtained materials including glass capillary (Kate experimental equipment business city company), Epoxy resin AB glue (Shanghai Shu Da chemical technology company), silicone tube and Polytetrafluoroethylene (PTFE) tubes (Kate experimental equipment business city company). Briefly, as external phase 1 and 2, two silicone tubes (inner diameter: 600 μm) were vertically inserted on the inside of PTFE tubes (inner diameter: 1mm), the outer phase 1 is 1 cm away from the top of the PTFE tubes, and the outer phase 2 is located in the middle of the PTFE tubes. Then, the internal phase glass capillary (inner diameter: 150 μm) was coaxially inserted into

the PTFE tubes, the inserted length was about 3cm. Afterwards, we used the Epoxy resin AB glue instantaneous adhesive to seal and fasten it. At last, we fixed the channel on the glass through the Epoxy resin AB glue.

Fabrication of no-woven fiber fabrics by MST.

First, BP-CNTs modified by 4-NBD was sonicated in N,N-dimethylacetamide (DMF), forming uniform dispersion. Then, a mixture of BP-CNTs dispersion, CNTs and TPU was further ultrasonic and stirred in DMF for 24 h at 60 °C, forming evenly dispersion. After that, a Y-shaped microreactor with three microchannels composing of one core flow (well dispersion of BP-CNTs, CNTs and TPU in DMF), and two sheath flow (C₂H₅OH and H₂O) was conducted. Typically, the core flow of dispersion (flow rate was 70 mL h⁻¹) was injected into the microreactor by syringe, followed by pumping the sheath flow of C₂H₅OH (external phase 1, flow rate was 55 mL h⁻¹) into device for pre-coagulation of dispersion. Besides, the dispersion was deep-coagulated to form microfibers by extruding water (external phase 2, flow rate was 40 mL h⁻¹) through solvent exchange. The as-prepared fibers were collected by filtration and further hot pressed and dried in the oven at 80 °C for 2 days, forming non-woven fiber fabrics. During the process of hot pressing, the non-woven fiber fusion was realized through heating or simple exposure to solvent vapours. The solvent-vapour-mediated allow multiple junctions of fibers and the fibers

could be interfused and interconnected with each other, forming non-woven fiber fabrics (Nature chemistry, 2009, 1:165-165, Adv. Mater. 17, 2177-2180).

Because of our carelessness, the picture of microfluidic chip device in Fig. 3a is not well matched with the real picture in Supplementary Fig. 4b.

Based on your advice, we have redrawn the microchannel in Fig. 3a.

The device that has three inlets, which matches well with Supplementary Fig. 4b. As seen SEM image in the inset of Fig.3a, the fabric consisted of uniform fibers with diameter of about 80 μm .

Figure 3 | Electrochemical performance of flexible SCs. (a) MST

fabrication of fibers-based no-woven fabrics. Left part is the illustration of microchannel of device. Middle part is the photo of no-woven fabric. The inset is the SEM image of fabric. Right part is the no-woven fabric that can be cut into various shapes and applied with different deformation. (b)~(d) Schematic illustration of the designed electrode structure. (e) CV curves of SCs at a scan rate of 10 mV s^{-1} . (f) Galvanostatic charge/discharge curves at the current density of 0.1 A cm^{-3} . (g) The calculated specific capacitances under different current densities. (h) EIS analysis of SCs. The inset figures are the depressed semicircle of Nyquist plots and the equivalent circuit model. Symbols denote experimental data, while the green lines represent the fitted data. (i) Cycle testing of SCs under a voltage of 3 V at a current density of 0.4 A cm^{-3} , the inset: Galvanostatic charge/discharge curves after 10,000 cycles. (j) Energy density versus power density of SCs compared with other electrodes based energy-storage systems. The used BP and BP-CNTs are all modified by 4-NBD.

The related revision is as follows:

“(On page 28) **Fabrication of microfluidic spinning device.** The microfluidic spinning device consists of two parts: syringe pumps (Nanjing Janus New Materials Co., Ltd) and triphase microfluidic chip. The microfluidic chip was fabricated with cheap and easily obtained materials including glass capillary (Kate experimental equipment

business city company), Epoxy resin AB glue (Shanghai Shu Da chemical technology company), silicone tube and Polytetrafluoroethylene (PTFE) tubes (Kate experimental equipment business city company). Briefly, as external phase 1 and 2, two silicone tubes (inner diameter: 600 μm) were vertically inserted on the inside of PTFE tubes (inner diameter: 1mm), the outer phase 1 is 1 cm away from the top of the PTFE tubes, and the outer phase 2 is located in the middle of the PTFE tubes. Then, the internal phase glass capillary (inner diameter: 150 μm) was coaxially inserted into the PTFE tubes, the inserted length was about 3cm. Afterwards, we used the Epoxy resin AB glue instantaneous adhesive to seal and fasten it. At last, we fixed the channel on the glass through the Epoxy resin AB glue.

Fabrication of no-woven fiber fabrics by MST. First, BP-CNTs modified by 4-NBD was sonicated in N, N-dimethylacetamide (DMF), forming uniform dispersion. Then, a mixture of BP-CNTs dispersion, CNTs and TPU was further ultrasonic and stirred in DMF for 24 h at 60 $^{\circ}\text{C}$, forming evenly dispersion. After that, a Y-shaped microreactor with three microchannels composing of one core flow (well dispersion of BP-CNTs, CNTs and TPU in DMF), and two sheath flow ($\text{C}_2\text{H}_5\text{OH}$ and H_2O) was conducted. Typically, the core flow of dispersion (flow rate was 70 mL h^{-1}) was injected into the microreactor by syringe, followed by pumping the sheath flow of $\text{C}_2\text{H}_5\text{OH}$ (external phase 1, flow rate was 55 mL h^{-1}) into device for pre-coagulation of dispersion. Besides, the

dispersion was deep-coagulated to form microfibers by extruding water (external phase 2, flow rate was 40 mL h⁻¹) through solvent exchange. The as-prepared fibers were collected by filtration and further hot pressed and dried in the oven at 80 °C for 2 days, forming non-woven fiber fabrics. During the process of hot pressing, the non-woven fiber fusion was realized through heating or simple exposure to solvent vapours. The solvent-vapour-mediation allows multiple junctions of fibers and the fibers could be interfused and interconnected with each other, forming non-woven fiber fabrics^{61,63}.

(On page 13) As illustrated in Fig. 3a, the microreactor with three microchannels is consisted of one core flow (well dispersion of 4-NBD modified BP-CNTs, CNTs and TPU), and two sheath flow (C₂H₅OH and H₂O).

(On page 13) Clearly, as seen SEM image in inset of Fig.3a, the fabric consists of uniform fibers with diameter of about 80 μm.”

-Water is added after ethanol, but there is nothing mentioned, where the ethanol remains. The authors state that water is extruded (l. 222) and the solvent is exchanged, but neither the extrusion nor the exchange of solvent is shown or proven, and I hardly believe that the ethanol “disappears”. Is it maybe diluted?

Reply: Thank you for the excellent review. Based on your review, we have done the experiments to demonstrate the solvent exchange of DMF

in ethanol and water. To highlight the experimental phenomena, we amplified the process by injecting well dispersion (BP-CNTs, CNTs and TPU in DMF solution) into ethanol and water. After solvent exchange, one thicker fiber can be formed. As clearly shown in Figure A, the fiber is pre-coagulated and the ethanol changed from clear and transparent to slightly turbid state, it is due to the exchange of solvents between ethanol and DMF. In addition, the same experimental operation is applied in the water system. Because of the exchange of solvents between H₂O and DMF, we can clearly see that the water has changed from clear to more turbid state. Thus, through solvent exchange, we obtain the fiber.

Figure A. The solvent exchange of DMF in ethanol and water.

To removal water and ethanol, the fiber-based no-woven fabric is dried and hot pressed in the oven at 80 °C for 2 days. As we know, the boiling points of ethanol and water are 78 °C and 100 °C, respectively. The water and ethanol will basically evaporated in the oven at 80 °C after 2 days. It

is worth noting that this hot pressing dried method has been reported by many Groups, such as Gao's group (Nature communication, 2016, 7: 13684) and Duan's group (Nature communication, 2014, 5: 4554). So, we consider that the water and ethanol can be removed through hot pressing dried in the oven at 80 °C for 2 days.

-The step from the microfluidic focusing to the formation of the macroscopic fabric with cm-dimensions is not clear (fig.3a and Fig. S4c), as well as the reason to produce fabrics of different forms (Fig 3a right image).

Reply: Thank you for the good review. Based on your review, we explain the formation of the macroscopic fabric with cm-dimensions. We first fabricate the fibers. A Y-shaped microreactor with three microchannels composing of one core flow (well dispersion of BP-CNTs, CNTs and TPU in DMF), and two sheath flow (C_2H_5OH and H_2O) was conducted. Typically, the core flow of dispersion (flow rate was 70 mL h^{-1}) was injected into the microreactor by syringe, followed by pumping the sheath flow of C_2H_5OH (external phase 1, flow rate was 55 mL h^{-1}) into device for pre-coagulation of dispersion. Besides, the dispersion was deep-coagulated to form microfibers by extruding water (external phase 2, flow rate was 40 mL h^{-1}) through solvent exchange. The as-prepared fibers were collected by filtration and further hot pressed and dried in the oven at 80 °C for 2 days, forming non-woven fiber fabrics. During the

process of hot pressing, the non-woven fiber fusion was realized through heating or simple exposure to solvent vapours. The solvent-vapour-mediated allow multiple junctions of fibers and the fibers could be interfused and interconnected with each other, forming non-woven fiber fabrics (Nature chemistry, 2009, 1:165-165, Adv. Mater. 17, 2177-2180, Nature communication, 2016, 7: 13684). In the preparation of fibers, we collect fibers and coagulant in a glass beaker. Then, the obtained fibers were filtered by vacuum filtration, dried and hot-pressed at 80 °C about 2 days (Nature communication, 2016, 7: 13684). Because of the solvent vapours, the fibers were connected into non-woven fabrics (Nature chemistry, 2009, 1:165-165). So, the size of the vacuum filtration determined the size of the nonwoven fabric.

According to your review, we explain how and why to produce fabrics of different forms. When obtaining the large size no-woven fabrics, we cut the fabrics into various shapes by a typical knife. It is the compact fabrics with strikingly mechanical strength that can be cut into various shapes and operated with different deformation, which ensure the robustness and flexibility of SCs. By cutting no-woven fiber fabrics into various shapes and bending test, we can demonstrate the flexibility and robustness of no-woven fabrics, this strategy has been reported by Gao's group (Nature communication, 2016, 7: 13684, Nature communication, 2011, 2: 571, Nature communication, 2014, 5: 3754), Chen's group

(Nature nanotechnology, 2014, 9:555-562) and Qu's group (Adv. Mater. 24, 1856-1861).

-The throughput seems to be low..?

In our work, the production speed of fiber is still a bit low. But, we will do our best to explore and improve our skills to realize the real faster and large-scale production of fibers. Based on previously reported work (Nature communication, 2018, 9:1222, Nature communication, 2016, 7: 13684), we believe that the large-scale preparation of microfibers through the MST within a short time will be realized soon. In the future, we have to achieve the large-area woven fiber-based supercapacitor to power mobile phone.

Reviewers' comments:

Reviewer #1 (Remarks to the Author):

The manuscript entitled "Microfluidic-Spinning-Construction of Black Phosphorous Hybrid Microfiber-Based Non-Woven Fabrics toward High Energy-Density Flexible Supercapacitor" presents non-woven fiber fabrics via microfluidic-spinning-technique of black phosphorous hybrid as wearable supercapacitor electrode materials. The flexible supercapacitor exhibits remarkably high energy - density, large volumetric capacitance, long life cycle stability and excellent deformable durability. In general, this work is interesting.

I recommend this manuscript to be published.

Reviewer #2 (Remarks to the Author):

I appreciate the improvements of the figure caption and the detailed description of the capillary device, but the changes are not implemented well in the manuscript. I think some questions are answered in the rebuttal letter but not changed in the manuscript. This should be overworked:

1) Concerning some former work on CNT/BP (now included as Ref. 74), I acknowledge the explanation why their work is innovative in the rebuttal letter. However, nothing of this is mentioned in the manuscript. I found the citation 74 at a somewhat hidden place on page 24.

2) My second question did not refer to microfluidic methods in general, but the question whether the technique is here in this study of benefit. It is still not clear to me.

3) Did the authors use two different device types? The glass capillaries, which is shown in the supplementary information as well as another one, that is not shown or further described and referred to as Y-shaped microreactor?

In this context, I believe the description needs some polishing in the language, e.g. "After that, a Y-shaped microreactor with three microchannels composing of one core flow (well dispersion of BP-CNTs, CNTs and TPU in DMF), and two sheath flow (C₂H₅OH and H₂O) was conducted."

I do not understand the design of this second reactor and believe the sentence is also wrong.

4) Figure 3a, this drawing is wrong. There is an inlet for the ethanol and another inlet for water. Both solutions will co-flow, there will be a gradient. In no way ethanol will just stop.

Response to reviewers' comments

Response to reviewer #1:

Comments:

The manuscript entitled "Microfluidic-Spinning-Construction of Black Phosphorous Hybrid Microfiber-Based Non-Woven Fabrics toward High Energy-Density Flexible Supercapacitor" presents non-woven fiber fabrics via microfluidic-spinning-technique of black phosphorous hybrid as wearable supercapacitor electrode materials. The flexible supercapacitor exhibits remarkably high energy-density, large volumetric capacitance, long life cycle stability and excellent deformable durability. In general, this work is interesting. I recommend this manuscript to be published.

Reply: Thanks very much for your approval of our manuscript.

Response to reviewer #2:

Comments:

I appreciate the improvements of the figure caption and the detailed description of the capillary device, but the changes are not implemented well in the manuscript. I think some questions are answered in the rebuttal letter but not changed in the manuscript. This should be overworked:

1) Concerning some former work on CNT/BP (now included as Ref. 74),

I acknowledge the explanation why their work is innovative in the rebuttal letter. However, nothing of this is mentioned in the manuscript. I found the citation 74 at a somewhat hidden place on page 24.

Reply: Thanks very much for your valuable suggestions. **According to your opinion, we have highlighted the work (ACS Appl. Mater. Interfaces 2017, 9, 44478-44484) in a prominent position of our manuscript, and we also have cited the relevant literatures.** As shown in the introduction at the end of page 4, underlying those unique properties, Liu's et al.⁵⁵ reported BP/CNTs hybrid materials through physical sonication, which endowed the supercapacitor with excellent electrochemical performance, including the specific capacitance of 41.1 F cm⁻³ and energy density of 5.71 mWh cm⁻³. However, poor electron conduction and stability between the BP interlayers severely limit their further improvement of energy-storage capability. In this work, considering from the chemical bonding design, we propose a hetero-structured BP-CNTs hybrid materials where 1D nanowire (CNTs) is chemical-bridged within 2D nanosheet (BP) via P-C bond connection under high-heat treatment. In such an architecture, BP shows a graphene-like conductivity. Thus, the CNTs in-situ embedded in BP flakes promote lamellar electron conduction, enhance mechanical stability and alleviate layers restacking of BP nanoflakes, which enable the conductive networks with more ionic channels (micro/meso-pores,

especially $< 1\text{nm}$) for ions fast diffusion and flooding storage.

To make our work more convincing, some related important references for characterizing the structures have been cited (Yang B-C, *et al.* Te-Doped Black Phosphorus Field-Effect Transistors. *Adv. Mater.* **28**, 9408–9415 (2016); Hao C-X, *et al.* Superior microwave absorption properties of ultralight reduced graphene oxide/black phosphorus aerogel. *Nanotechnology* **29**, 235604 (2018)).

The related revision is as follows:

“On page 4

Underlying those unique properties, Liu’s *et al.*⁵⁵ reported BP/CNTs hybrid materials through physical sonication, which endowed the supercapacitor with excellent electrochemical performance, including the specific capacitance of 41.1 F cm^{-3} and energy density of 5.71 mWh cm^{-3} . However, poor electron conduction and stability between the BP interlayers severely limit their further improvement of energy-storage capability.

In this work, considering from the chemical bonding design, we propose a hetero-structured BP-CNTs hybrid materials where 1D nanowire (CNTs) is chemical-bridged within 2D nanosheet (BP) via P-C bond connection under high-heat treatment. In such an architecture, BP shows a graphene-like conductivity. Thus, the CNTs in-situ embedded in BP flakes promote lamellar electron conduction, enhance mechanical

stability and alleviate layers restacking of BP nanoflakes, which enable the conductive networks with more ionic channels (micro/meso-pores, especially < 1nm) for ions fast diffusion and flooding storage.

One page 35

65. Yang B-C, *et al.* Te-Doped Black Phosphorus Field-Effect Transistors. *Adv. Mater.* **28**, 9408–9415 (2016).

66. Hao C-X, *et al.* Superior microwave absorption properties of ultralight reduced graphene oxide/black phosphorus aerogel. *Nanotechnology* **29**, 235604 (2018).”

2) My second question did not refer to microfluidic methods in general, but the question whether the technique is here in this study of benefit. It is still not clear to me.

Reply: Thank you for the nice suggestion. Depending on your opinion, we have described the beneficial of microfluidic methods in our study. **There are two advantages of microfluidic fabrication. 1. Microfluidic-spinning-technique (MST) that can fabricate extensible fiber-based fabric electrodes, enables the electrode with excellent mechanical strength such as flexibility and deformability.** The as-prepared fabric electrodes via MST have shown outstanding mechanical elongation of 17.96%, which is much higher than that of other literatures reported methods (6.42 % in *Advanced Functional Materials*. 2017, 27, 1606219; 5.8 % in *Nature Commutations*, 2011, 2,

571; 6% in *Advanced Materials* 2012, 24, 1856–1861, 10% in *Nature Communications*, 2014, 5, 3754). **2. Compared with fiber-based supercapacitors, MST can enable fibers assembly into non-woven fabric, which largely improves the output work energy.** The energy density of our supercapacitor is 96.5 mWh cm^{-3} , which is higher than that of carbon materials-based fiber supercapacitors (2.5 mWh cm^{-3} of rGO (*Nature Communications*, 2013, 4, 2487); 6.3 mWh cm^{-3} of CNTs/rGO (*Nat Nanotechnol*, 2014, 9, 555-562)). As a result, our fabric-based supercapacitors can remarkably power various electronics with both low and high start-up currents, including LEDs, smart watch and displays, whereas other supercapacitors are basically limited to power low-current-driven LEDs.

The related revision is as follows:

“On page 13

To improve the mechanical properties, fabrics were compressed to form compact films with excellent electrical conductivity ($75.2 \text{ } \Omega/\square$) and mechanical strength (Young’s modulus, 313 MPa; break elongation, 17.96% (Supplementary Fig. 6). **It is worth mentioning that the mechanical elongation of fabric electrodes by MST is much higher than that of other literatures reported electrode materials (Graphene chiral liquid crystals electrode, 5.8%)⁶⁹, (Graphene fibers, 6%)⁷⁰, (CNTs/ZIF-8 film, 6.42%)²⁵, (RGO+CNTs@CMC, 10%)¹⁸, ensuring the excellent**

flexibility and deformability of SCs.

On page 25

Secondly, we develop MST to scalable fabrication of flexible electrodes. In particular, microfibers prepared by MST are assembled into no-woven fiber fabrics. As a result of fibers assembly, the nanoscale effects of individual microfiber, including unique porous structure, electrochemical activity, electrical conductivity and flexibility are amplified to integrate on the whole no-woven fabrics. Accordingly, no-woven film based flexible SC displays high overall capacitance and energy density for powering various electronic devices.”

3) Did the authors use two different device types? The glass capillaries, which is shown in the supplementary information as well as another one, that is not shown or further described and referred to as Y-shaped microreactor? In this context, I believe the description needs some polishing in the language, e.g. “After that, a Y-shaped microreactor with three microchannels composing of one core flow (well dispersion of BP-CNTs, CNTs and TPU in DMF), and two sheath flow (C₂H₅OH and H₂O) was conducted.” I do not understand the design of this second reactor and believe the sentence is also wrong.

Reply: Thank you for the good suggestion. As you mentioned, we have used one type of device. It is our carelessness that we have not describe our device properly in English writing. Under your suggestion, we have

revised the wrong description of “Y-shaped microreactor” into “triphase microfluidic microreactor”. We also have polished our language for more understandable.

The related revision as follows:

On page 30:

“After that, a triphase microfluidic microreactor with three microchannels, which is composed of one inner core flow (well dispersion of BP-CNTs, CNTs and TPU in DMF), and two outer sheath flows (C_2H_5OH and H_2O , respectively) was conducted. Typically, the inner core flow of dispersion (flow rate was 70 mL h^{-1}) was firstly injected into the microreactor by a syringe, followed by pumping the outer sheath flow of C_2H_5OH (external phase 1, flow rate was 55 mL h^{-1}) into device for pre-coagulation of dispersion. After a while, the outer sheath flow of H_2O (external phase 2, flow rate was 40 mL h^{-1}) was further injected for deep-coagulation, forming microfibers through solvent exchange.”

On page 13: We have revised “syringe” into “a syringe”.

On page 13: We have revised “It is because” into “Since”.

On page 13: We have revised “so that” into “these”.

On page 13: We have revised “interfused” into “also interfused”.

4) Figure 3a, this drawing is wrong. There is an inlet for the ethanol and another inlet for water. Both solutions will co-flow, there will be a

gradient. In no way ethanol will just stop.

Reply: Thanks for your kindly suggestion. You point out the wrong drawing in our manuscript. Based on your advice, we have redrawn the schematic illustration of triphase microfluidic microreactor. As described in Figure 3a, there are two sheath inlets: one is for the ethanol and the other is for water. Because both solutions will co-flow, there will be a gradient at the outlet, which now is well illustrated in Figure 3a.

The related revision is on the right of page 20:

Figure 3 | Electrochemical performance of flexible SCs. (a) MST fabrication of fibers-based no-woven fabrics. Left part is the illustration

of microchannel of device. Middle part is the photo of no-woven fabric. The inset is the SEM image of fabric, scale bar 500 μm . Right part is the no-woven fabric that can be cut into various shapes (scale bar 1 cm) and applied with different deformation (scale bar 2 cm). **(b)~(d)** Schematic illustration of the designed electrode structure. **(e)** CV curves of SCs at a scan rate of 10 mV s^{-1} . **(f)** Galvanostatic charge/discharge curves at the current density of 0.1 A cm^{-3} . **(g)** The calculated specific capacitances under different current densities. **(h)** EIS analysis of SCs. The inset figures are the depressed semicircle of Nyquist plots and the equivalent circuit model. Symbols denote experimental data, while the green lines represent the fitted data. **(i)** Cycle testing of SCs under a voltage of 3 V at a current density of 0.4 A cm^{-3} , the inset: Galvanostatic charge/discharge curves after 10,000 cycles. **(j)** Energy density versus power density of SCs compared with other electrodes based energy-storage systems. The used BP and BP-CNTs are all modified by 4-NBD.